# Impacts of hot-dry compound extremes on US soybean yields

Raed Hamed[1], Anne F. Van Loon[1], Jeroen Aerts[1,2], Dim Coumou[1,3]

[1] Department of Water and Climate Risk, Institute for Environmental Studies (IVM), Vrije Universiteit Amsterdam, Amsterdam, the Netherlands
[2] Deltares Institute, Delft, The Netherlands
[3] Royal Netherlands Meteorological Institute (KNMI), De Bilt, the Netherlands

*Correspondence to*: Raed Hamed (raed.hamed@vu.nl)

**Abstract.** The US agriculture system supplies more than one-third of globally-traded soybean and with 90% of US soybean produced under rainfed agriculture, soybean trade is particularly sensitive to weather and climate variability. Average growing season climate conditions can explain about one-third of US soybean yield variability. Additionally, crops can be sensitive to specific short-term weather extremes, occurring in isolation or compounding at key moments throughout crop development. Here, we identify the dominant within-season climate drivers that can explain soybean yield variability in the US, and explore synergistic effects between drivers that can lead to severe impacts. The study combines weather data from reanalysis and satellite-informed root-zone soil moisture fields with sub-national crop yields using statistical methods that account for interaction effects. Our models can explain on average about two thirds of the year-to-year yield variability (70% on all years and 60% on out-of-sample predictions). The largest negative influence on soybean yields is driven by high temperature and low soil moisture during the summer crop reproductive period. Moreover, due to synergistic effects, heat is considerably more damaging to soybean crops during dry conditions, and less so during wet conditions. Compound and interacting hot and dry summer conditions (defined by the 95th and 5th percentiles of temperature and soil moisture, respectively) reduce yields by 2 standard deviation. This sensitivity is, respectively, 4 and 3 times larger than the sensitivity to hot or dry conditions alone. Other relevant drivers of negative yield responses are lower temperatures early and late in the season, excessive precipitation in early season and dry conditions in late season. The sensitivity to the identified drivers varies across the spatial domain with higher latitudes, and thus colder regions, positively affected by high temperature during the summer period. On the other hand, warmer south-eastern regions are positively affected by low temperature late season. Historic trends in identified drivers indicates that US soybean has generally benefited from recent shifts in weather except for increasing rainfall in the early season. Overall warming conditions have reduced the risk of frost in early and late-season and potentially allowed for earlier sowing dates. More importantly, summers have been getting cooler and wetter over eastern US. Still, despite these positive changes, we show that the frequency of compound hot-dry summer events has remained unchanged over 1946-2016. In the longer term, climate models project substantially warmer summers for the continental US but uncertainty remains whether this will be accompanied by drier conditions. This highlights a critical element to explore in future studies focused on US agricultural production risk under climate change.

## 1 Introduction

Soybean is one of the most in-demand crops worldwide, with the largest increases in production-area over the last two decades when compared to all other major staple crops (Hartman et al., 2011). A considerably large portion of this production is dedicated to animal feed accommodating the current global increase in demand for animal products (Cassidy et al., 2013). A recent estimate based on FAOSTAT data in 2013 reports that soybean ranks second in terms of globally-produced kilocalories (~20% of the total kcal traded on the global food market) and first among staple crops in terms of globally-aggregated trade monetary value (Torreggiani et al., 2018). The US agriculture system alone supplies more than one-third of globally-traded soybean, of which 90% is produced under rainfed agriculture (Jin et al., 2017). The recent surge in global soybean demands is expected to increase further in the future due to increasing global population and associated shifts in dietary preferences (Fehlenberg et al., 2017). At the same time, climate change is expected to increase annual mean and extreme temperature levels over the US (Dirmeyer et al., 2013; Winter et al., 2015; Wuebbles et al., 2014a). To support adaptation measures that reduce the potential impacts of these future challenges, we need a quantitative understanding of crop sensitivity to climate and weather variables.

Climate variability can strongly impact crop yields. The effects of growing season temperature and precipitation conditions can explain about one-third of US soybean year-to-year yield variability (Leng et al., 2016; Lobell et al., 2011; Ray et al., 2015; Vogel et al., 2019). In particular, heat and drought conditions are among the most limiting environmental factors affecting crops (Lesk et al., 2016). These are increasingly detrimental when coinciding with vulnerable stages of the crop growth cycle (Troy et al., 2015). Such conditions can occur separately or in combination, in the latter case, leading often to more severe impacts (Leonard et al., 2014). For instance, it is reported that US economic agricultural losses between 1980 and 2012 are four times larger during hot and dry conditions compared to drought events alone (Suzuki et al., 2014). Moreover, the response to multiple climatic stressors is complex and can be subject to interaction effects where climatic drivers create more damage in combination than the sum of each in isolation (Ben-Ari et al., 2018; Haqiqi et al., 2021; Matiu et al., 2017; Rigden et al., 2020). Interestingly, multiple climatic stressors can also result in positive interactions with beneficial effects on crop yields (Carter et al., 2016; Suzuki et al., 2014). Such features, positive or negative, are likely to have important implications on future impacts and adaptation strategies to climate change. Nevertheless, these have received little attention in current assessments so far (Matiu et al., 2017; Zscheischler et al., 2017).

A compound event framework has lately been proposed to underline the need for impact-centric approaches that identify multiple climatic drivers contributing to socio-economic risk (Leonard et al., 2014; Zscheischler et al., 2018, 2020). The types of damaging combination of drivers on local agricultural production are various, with a specific terminology recently proposed in Zscheischler et al. (2020). These can be temporally compounding, as in the case of the 2016 wheat production in France where high temperatures during winter followed by heavy precipitation during spring lead to unprecedented yield losses (Ben-

Ari et al., 2018). These can be preconditioned where for instance, pre-sowing soil moisture water storage content interacts with within-season precipitation to affect rainfed maize yield in the US (Carter et al., 2018a) or multivariate/co-occurring such as in the case of hot-dry conditions in the growing season affecting crop yields (Feng and Hao, 2020; Matiu et al., 2017). One way to identify such drivers is through the use of statistical methods that empirically associate drivers to impacts (Vogel et al., 2021). Easily interpretable linear regressions in that context can be useful tools, in particular when fitted with alternative methods that allow for the consideration of a large number of potential predictors (i.e. subset selection, shrinkage or dimension reduction approaches) (Ben-Ari et al., 2018; Carter et al., 2018a; Laudien et al., 2020; Vogel et al., 2021).

Here we analyze soybean yields and climate time series for the U.S. at the county scale from 1982 to 2016 using regression models that are fitted with a reduced set of variables selected via a subset selection approach. The aim is to identify (1) the combination of climatic conditions affecting soybean yields at different stages of the growing season, and (2) potential interaction effects between drivers modulating the final impact on yield. Furthermore, we study (3) trends in the identified dominant climate drivers from 1946 to 2016 to assess how historic trends likely affected soybean production risk. Finally, we explore (4) how temperature and moisture couplings differ within the growing season between hot-dry summers and normal summers. We discuss how that potentially affects the occurrence of compound hot-dry extremes and associated crop impacts.

## 2 Data and Methods

### 2.1 Soybean yields, climate and hydrological data for the U.S.

Soybean yields are analysed at the county scale for the period 1982-2016, based on census data obtained from the US Department of Agriculture (USDA) National Agriculture Statistics Survey (NASS) Quick Stats database (www.nass.usda.gov/Quick_Stats). Counties are selected on (i) having no missing data for the full 35 years analysed, (ii) have common planting dates (i.e. April-May) and (iii) a production area share of at least 90% rainfed agriculture. Consequently, a total of 389 counties are retained for the regression analysis (Fig. 1). These together account for at least 50% of US total rainfed soy production, where production per county is calculated as the average production over 1982-2016. In the study region, planting dates are aligned to provide comparable crop growth stages between counties. This facilitates the interpretation of climate sensitivities  associated to timing within the growing season. Information on the soybean growing season and rainfed vs irrigated agricultural land cover is obtained from the monthly irrigated and rainfed crop areas database around the year 2000 (MIRCA2000), a global gridded dataset at 0.5° resolution (Portmann et al., 2010). The percent rainfed area is calculated by dividing the rainfed area in each grid cell by the total harvested area for each cell (Schauberger et al., 2017a). A linear trend is removed from yield values at the county scale to eliminate long-term effects largely due to technological improvements over the study period (Fig. S1) (Li et al., 2019; Zipper et al., 2016).

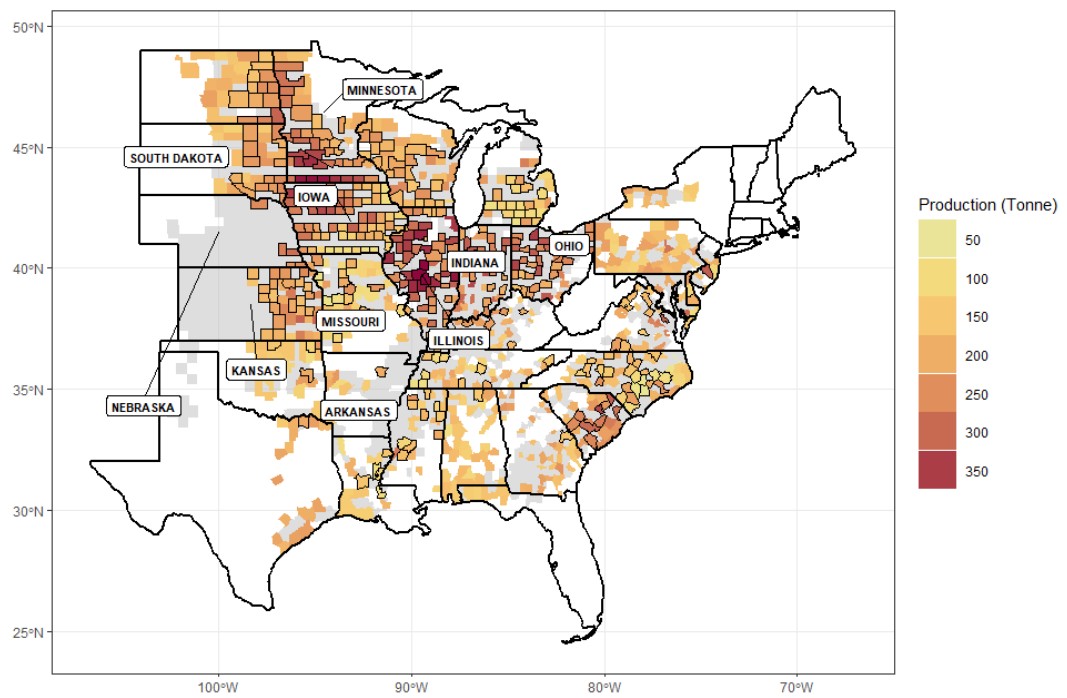

**Figure 1. Average total production in tonnes over the period of study (1982-2016). Counties with 35 years of data are highlighted with a thin black perimeter. Grey regions represent filtered out counties where local agriculture is less than 90% rainfed.**

Global hydrological and weather datasets are used for this analysis. This provides the possibility to conduct similar assessments, in other parts of the world, whenever impact data is available. Nevertheless, other studies can benefit from leveraging local climate and hydrological data when available for better representativeness. Root zone soil moisture (SMroot) variable ($m^3/m^3$) is obtained from the modelled GLEAM v3.3a dataset that incorporates an observed satellite-based soil moisture data assimilation system (Martens et al., 2017). The dataset is available at a 0.25° grid resolution and a daily time step that covers the period of study (1982-2016). Weather data, namely maximum (Tmax) and minimum (Tmin) temperature (°C) in addition to precipitation (mm) are obtained from the bias-adjusted WFDE5 reanalysis covering the same period (1982-2016) at daily time step and a 0.5° grid resolution (Cucchi et al., 2020). Daily precipitation is further processed into number of days with precipitation above 20 mm (Num_pr20) to explicitly account for potential negative effects of excessive precipitation on yield (Li et al., 2019; Zhu and Troy, 2018). All variables are temporally aggregated to monthly and seasonal windows over early- (April-May), mid- (June-July-August) and late-growing season (September-October) periods. Additionally, variables are spatially aggregated to the county scale based on county boundary maps of the 2016 US Census Bureau. A summary of the considered variables for the modelling analysis is presented in Table 1. Dividing the growing season by calendar months allowed the identification of key phases throughout the season where soybean crops are most sensitive to climate variability. These can reflect both vulnerable physiological crop growth stages and important climatic thresholds. We

could have used a more complex characterization of crop developmental stages based on phenological heat units (Schauberger et al., 2017b) or the consideration of sub-monthly aggregation periods for climatic time series, but these did not necessarily improve model performance in other assessments and therefore we opted here to simply rely on monthly and seasonal estimates (Ben-Ari et al., 2016; Ortiz-Bobea et al., 2019; Sharif et al., 2017). Full growing season averages have been tested as potential predictors but these did not improve modelling results and have therefore been omitted from further analysis. We thus exclusively focus on within season crop climate sensitivities.

**Table 1. Climate variables calculated at seasonal and monthly time scales throughout the growing season**

| Variable abbreviation | Variable explanation | Type | Unit |
| --- | --- | --- | --- |
| **Tmin** | Average minimum Temperature | Temperature related | °C |
| **Tmax** | Average maximum Temperature | Temperature related | °C |
| **Num_pr20** | Number of days with precipitation above 20 mm | Moisture related | days |
| **SMroot** | Root zone soil moisture | Moisture related | $m^3/m^3$ |

## 2.2 Simulating yield variability

We used regression models to estimate yield variability at the county scale. Typically, three types of statistical models are used in such assessments (i.e. time-series, panel, and cross-sectional models) (Lobell and Burke, 2010). Here we opted for time-series models as these are (i) easy to interpret, (ii) often perform well compared to the other approaches, and (iii) allow for spatially heterogeneous parameter estimation that may highlight important local and regional features (Gornott and Wechsung, 2016). Out of all possible models constructed with a single input variable at county scale, we selected the most influential moisture- and temperature-related variables per county based on the Bayesian Information Criterion (BIC) (Ben-Ari et al., 2018). This was done separately for early- (April-May), mid- (June-July-August) and late-growing season (September-October) periods considering both monthly and seasonal aggregates for each, and thus, ended up with a subset of six best predictors for each county. Finally, we applied a stepwise selection procedure to identify the best combination of these input variables, with and without interactions, picking the model with the lowest BIC value at county level (Ben-Ari et al., 2018). The stepwise approach considers all selected variables and all possible interactions (i.e. products of all possible pairs of selected predictors). The procedure is then to start from a model with no predictors, sequentially adding and removing predictors until only a subset is left resulting in the most parsimonious model with the lowest prediction error on training data (See step.lm function of R, version 3.6.1). The performance of the resulting model was evaluated using the coefficient of determination ($R^2$). Further robustness tests with respect to both predictor selection and model performance are detailed in the following subsection. A summary of the modelling framework is presented in Fig. 2.

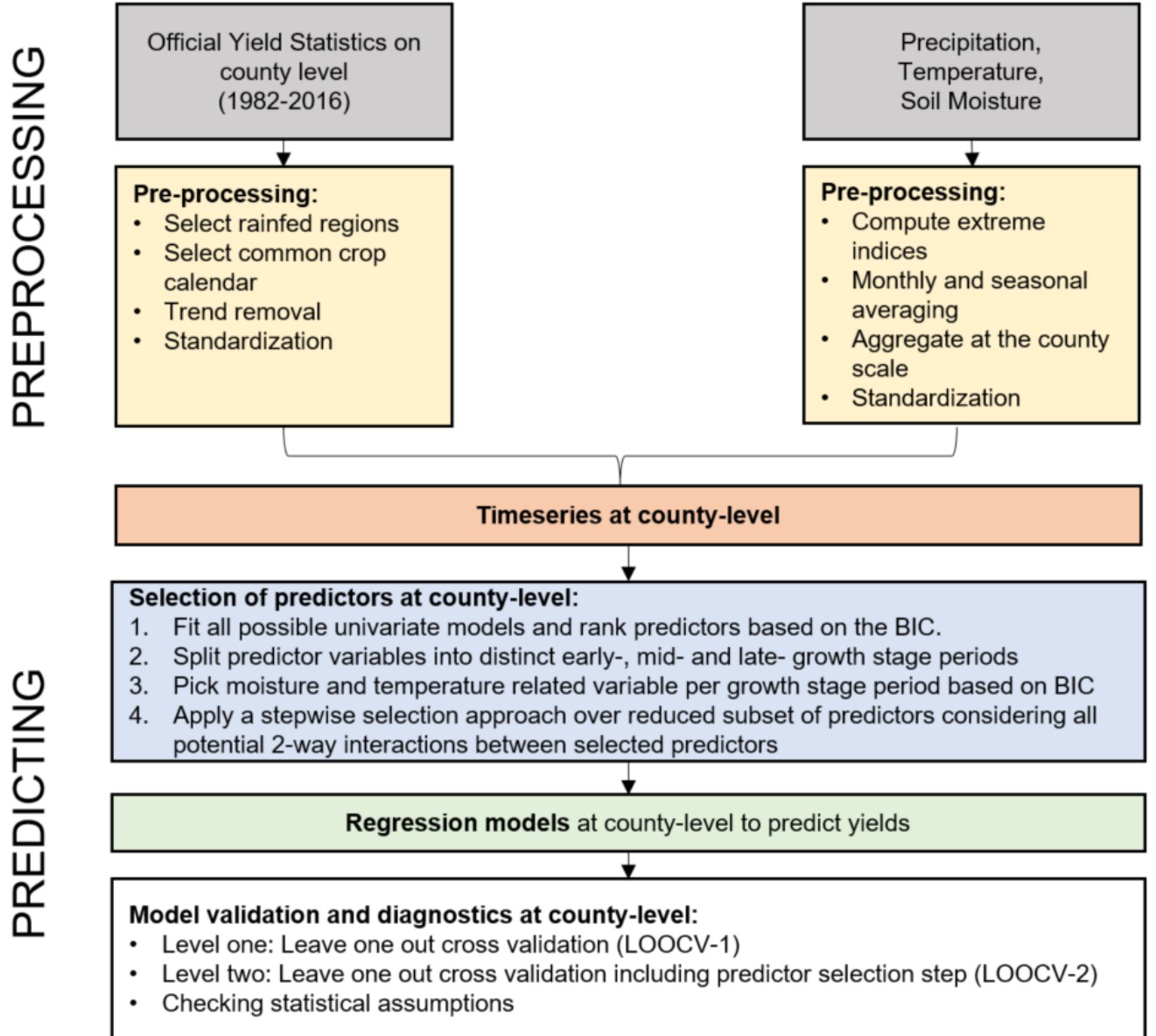

**Figure 2. Overall modelling workflow applied for this study linking US yields to weather and climate variables.**

### 2.3 Validating performance and testing modelling assumptions

To test robustness of the model performance and the selected predictors, we applied a two level leave-one-out cross-validation scheme (LOOCV) (Laudien et al., 2020). Level one (LOOCV-1) consisted of training county-scale models on reduced datasets. These are constructed by iteratively removing the to-be-forecasted year and predicting the one out of sample value using a set of predictors per county selected using the complete dataset. Level two (LOOCV-2) is similar but repeats the predictor selection

step for every iteration. This way, we completely eliminate information shared between training and validation sets. Furthermore, we calculated how often selected predictors are chosen across each iteration in the cross-validation procedure of LOOCV-2. Both elements, respectively, provide a more robust model performance estimate and predictor selection step. The adequacy of applying linear models at the county scale for assessing the relationship between yield anomalies and selected predictors was successfully assessed using five statistical tests (Gornott and Wechsung, 2016; Schauberger et al., 2017b). The regression equation specification error test (RESET) assessed whether taking powers of the predictor variables would improve the model fit. The Breusch-Pagan test examined heteroscedasticity issues with the data. The Breusch–Godfrey test was used to assess autocorrelation and the Shapiro–Wilk test to examine normality of residuals. Multicollinearity was checked using the variance inflation factor calculated for each independent variable while setting acceptable levels to strictly below 3.

## 2.4 Changes in key climatic conditions from 1946 to 2016

Historic trends of the dominant climatic drivers were assessed for the period 1946 to 2016 using linear regressions (0.05 significance level). Furthermore, we assessed changes in concurrent hot-dry summer conditions as these were shown to be particularly relevant for soybean production. The selected input datasets used in the crop-modelling analysis do not cover years preceding 1981. To overcome this limitation, we used precipitation, number of wet days and temperature minimum and maximum variables from the CRU V4 global dataset (Harris et al., 2020) covering the period 1901-2019 at a spatial resolution of 0.5°. Number of wet days in the early season was used as a proxy for early season number of days with precipitation above 20 mm. Mean summer precipitation over June-July-August-September was used as a proxy for August-September averaged root zone soil moisture. To check the feasibility of these assumptions, we calculated correlation maps between GLEAM August-September averaged root zone soil moisture and CRU averaged summer precipitation and between WFDE5 spring number of days with precipitation above 20 mm and CRU spring number of wet days for the period 1982 to 2016. The mean Pearson's correlation coefficient over the whole spatial domain was 0.66 for summer precipitation and root zone soil moisture and 0.83 for spring number of wet days and number of wet days above 20 mm (Fig. S2). The $25^{th}/10^{th}$ and $75^{th}/90^{th}$ percentiles of summer precipitation and August maximum temperature are used to jointly define the compound hot-dry events at the local scale. Accordingly, we calculated the percent-change per grid cell based on the difference between the number of compound events over two distinct periods (1946-1980 relative to 1982-2016) normalized by the total amount of events over the entire analysis period. Statistical significance of this percent change is assessed using the non-parametric Wilcoxon Rank Sum test (0.05 significance level). Moreover, we calculated a percent (%) area time series of the total rainfed producing region under compound summer hot-dry conditions by summing the number of grid cells under such conditions for a given year and dividing by the total number of grid cells considered, similar to the approach applied in Mazdiyasni and AghaKouchak (2015). The trend in the aforementioned time-series was assessed with the non-parametric Mann−Kendall trend test (0.05 significance level).

## 2.5 Exploring temperature and moisture couplings during summer hot-dry events

To get insight on how key elements related to moisture and temperature couplings differ during compound summer hot-dry years, we estimated the coevolution of actual evapotranspiration, root zone soil moisture and maximum temperature pairs composited into hot-dry events for the period 1982-2016. Hot-dry summer events in this case are defined as years when more than 20% of the total harvested area is under hot-dry conditions (using the 75th and 25th percentiles respectively). Coevolution of considered variables was estimated by calculating the interannual correlation between pairs of variables for a given month of the year, repeated over the various calendar months (Seneviratne et al., 2010). This calculation incorporates data from all counties into the correlation by first spatially averaging single variables over the entire rainfed harvested area (Fig. 1) and then quantifying the couplings. Moreover, we calculated correlation at the grid-cell level between actual evapotranspiration and maximum temperature to show how these couplings can differ at the local scale. Actual evapotranspiration (AET) (mm) is retrieved from the GLEAM v3.3a dataset with the same temporal and spatial resolution of aforementioned root zone soil moisture variable. AET within the GLEAM dataset is derived from potential evapotranspiration model estimates multiplied by an evaporative stress factor based on observations of microwave vegetation optical depth (VOD) and root zone soil moisture values.

## 3 Results

### 3.1 Overall model performance

Based on the selection procedure shown in Fig. 2, we identify a set of non-redundant moisture and temperature variables at different stages of the growing season that can best explain yield variability at county scale. These varied across the spatial domain (Fig. A1, A2) with dominant patterns summarized as follows: Excessive precipitation is highlighted as the main driver of reduced soybean yields in the early season alongside low minimum and maximum temperature values. Low soil moisture and high maximum temperature values are highlighted as main drivers of reduced yields in the mid-season, particularly for the months of August and September. Finally, low soil moisture and low minimum temperature values are highlighted as main drivers of reduced yields late in the season (Fig. 3a). The trained regression models at county level with identified predictors are able to explain, on average, about two-thirds of the year-to-year yield variability (70% on all years and 60% on LOOCV-1 predictions). Including interaction terms in the fitted model contributed to 10% out of the total 60% explained variability on LOOCV-1 predictions. Testing the model with the more conservative LOOCV-2, repeating the predictor selection step at every iteration, lowers model explained variability to 30% (Fig. 3a). This reduced performance is expected when comparing with results of studies that applied a similar robust leave one out cross-validation approach (Laudien et al., 2020; Lehmann et al., 2020). Still, for ~83% of the years, the LOOCV-2 model provides a correct year-to-year direction of change as well as sign of the yield anomaly (i.e. positive or negative) (Fig. 3b). Furthermore, most frequently selected predictors and associated timing within the season across the training sets shows high consistency and good agreement with predictors selected on the full

dataset (Fig. S3, S4, S5). This provides confidence with respect to the choice of predictors. Overall, the dominant crop yield drivers are August/September root zone soil moisture and August maximum temperature, each selected over more than 25% of considered counties. Averaged standardized beta coefficients for aforementioned variables reports the highest absolute value of around 0.4 (i.e. ~0.4 standard deviation change in soybean yields per standard deviation change in the predictor when excluding the effect of interaction terms). Furthermore, interaction effects between summer moisture and temperature variables are the most frequently selected type of interaction (Fig. A3).

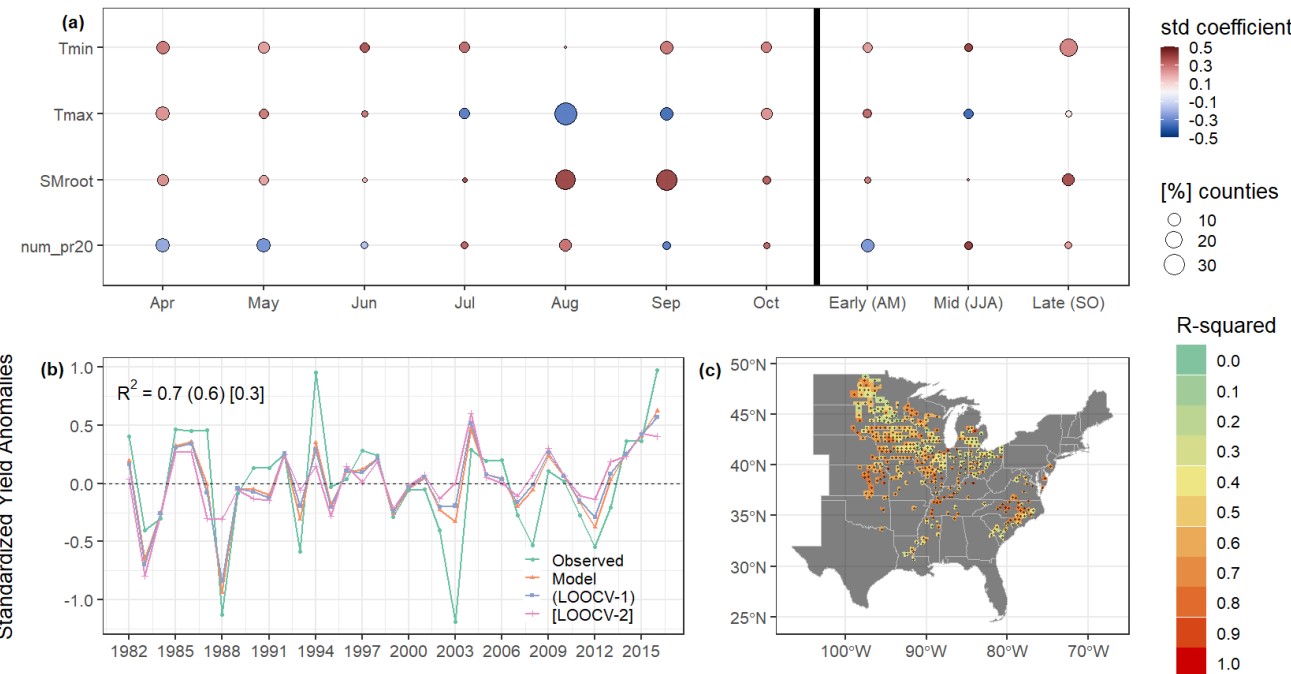

**Figure 3. (a) summary of the strength and frequency of selected predictors across the growing season. (b-c) Explained variance (R-squared) of yield anomalies due to climate variability (b) spatially averaged and (c) at the county scale. Stippling in (c) shows F-tests with (p < 0.05) indicating that the model chosen is significantly better than a null model (accounting for false discovery rate due to multiple hypotheses testing).**

Spatially, the model is statistically significant (p-value < 0.05) for all considered counties (Fig. 3c) after adjusting for multiple hypotheses testing using the False Discovery Rate (FDR) method (Ventura et al., 2004). Yield variability is captured particularly well in southern counties (Fig. 3c), with high performance represented by red shading ($R^2$ ~ 0.8). On the other hand, the model performs slightly poorer in northern counties, consistent with the results of Schauberger et al. (Schauberger et al., 2017b) where regional colder and wetter climatology reduces soybean yield sensitivity to hot-dry conditions. Individual diagnostic tests for models built at the county scale shows that autocorrelation and heteroscedasticity did not occur for the majority of individual models whereas model residuals are mostly normally distributed. The RESET test shows that most models are properly specified meaning that considering quadratic variables would not have improved the model fit. Although quadratic associations between crop yields and climatic variables are well established, these often are highlighted for

seasonally averaged temperature and moisture conditions (Ray et al., 2015). Dividing the growing season into smaller periods in this study likely made these non-linear associations less relevant. Finally, the VIF value is strictly smaller than 3 for the majority of considered models and variables reflecting low multicollinearity concerns (Fig. A4).

## 3.2 Spatial variability of model coefficients

The spatial variability of crop yield sensitivities to the selected predictors is depicted in Fig. 4a-f. It shows county-based standardized model coefficients and associated patterns across the spatial domain for both moisture and temperature related variables and for early, mid, and late season. Specifically selected predictors and associated timing within the season per county are shown in Fig. A1 & A2.

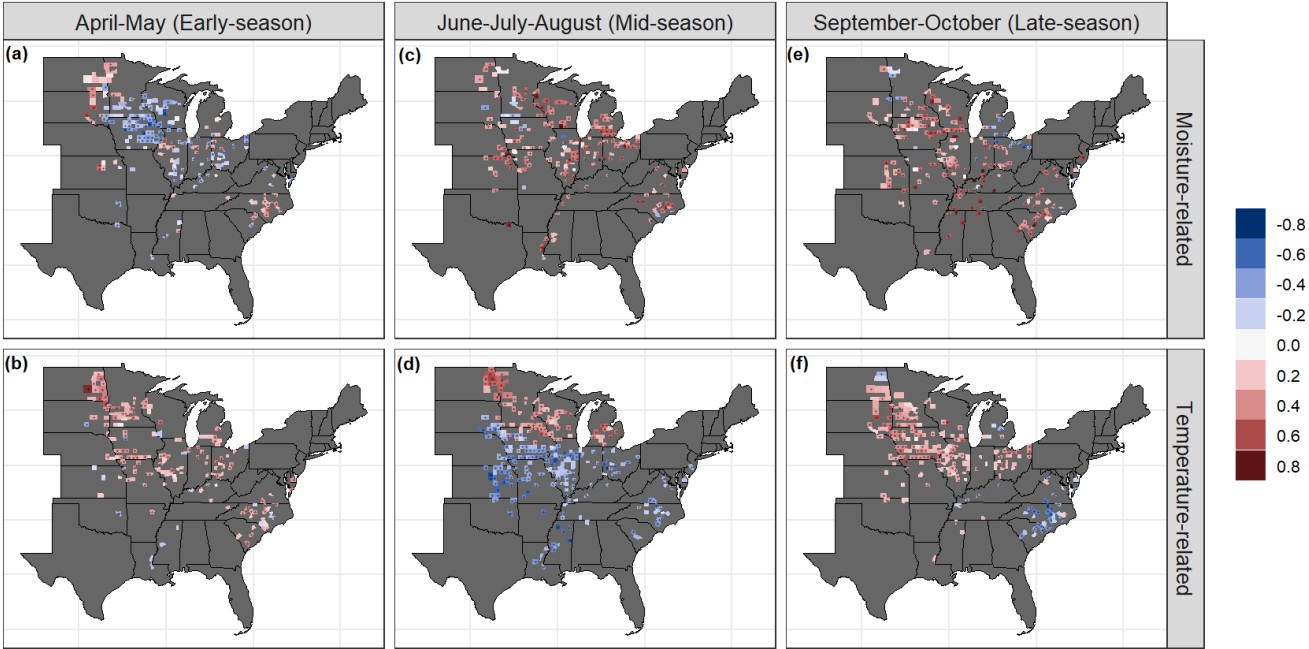

**Figure 4. Region- and season-specific estimated sensitivity coefficients for soybean yield and selected predictors. Stippling indicates statistical significance from a t-test at 95% confidence level. Values of coefficients are interpreted as the change in soybean yield standard deviation from a one-standard deviation change in the considered independent variable. Temperature-related variables can refer to either Tmin or Tmax depending on the selected variable in a given county. Similarly, moisture-related variable can refer to either SMroot or Num_pr20. Finally, for each seasonal bracket (i.e. Early, Mid or Late), the selected time resolution for each variable can be either a seasonal aggregate or the value for a specific month within that bracket. We refer the reader to the appendix (Fig. A1, A2) for a more detailed account of selected variables per county.**

Early season reports mainly a negative relationship between yield and moisture variables (Fig. 4a) across the majority of the spatial domain in line with Ortiz-Bobea et al., (2019). The most frequently selected predictor is number of days with precipitation above 20 mm used as a proxy for excessive rain (Fig. 3a, A1). The signal is particularly strong and significant near Iowa and Minnesota where soils are generally poorly drained (i.e. high clay fraction, low saturated hydraulic conductivity) (Li et al., 2019). The temperature related variable in early season (Fig. 4b) shows a positive relationship with yields, and this

can reflect both minimum and maximum temperature (Fig. A1). During the mid-season, temperature-related variables negatively affect soybean yields across the spatial domain. Exceptions are for northern states (north of Iowa and Illinois) where the sensitivity is reversed and higher temperature lead to positive effects on yield (Fig. 4d). The selected variable for the negative sensitivity (for southern states) refers mostly to maximum temperature in August whereas the positive sensitivity (for northern states) refers mostly to minimum temperature in June and July (Fig. A1, A2). Moisture related variables have a strong positive influence on yields both in the mid and late season (Fig. 4e). In particular, selected predictors are predominantly soil moisture variables in August and September. Temperature sensitivities in the late season show mostly positive effects on yield, except for counties in south-eastern states which show strong negative sensitivities (Fig. 4f). The selected late-season temperature predictor is predominantly minimum temperature for the positive associations and September maximum temperature for the negative associations over southern states (Fig. A1, A2). Furthermore, interaction terms between summer soil moisture and temperature variables are included in ~10% of the considered counties across the spatial domain (Fig. A3). These interaction effects imply that the impact of summer temperature on crop yields significantly depends on the concurrent soil moisture levels in those areas. The negative effects of high temperatures are amplified during dry conditions and alleviated during wet conditions (see Sect. 3.3). Moreover, another interaction term is picked up, albeit less pronounced, between maximum August temperature and end of season minimum temperature mostly within Iowa (Fig. A3). This might reflect increased impacts whenever anomalously hot conditions in peak summer are followed by anomalously cold conditions in September-October. Optimal temperature for crop photosynthesis fluctuates due to the capacity of the crop to seasonally adjust its physiological response to temperature (Kumarathunge et al., 2019). It follows that consistent high temperature within the growing season can make crops more productive at higher temperatures. The abrupt change in temperature conditions from hot to cold further stresses crops and reduces the potential positive effects of crop temperature acclimation (Butler and Huybers, 2013; Carter et al., 2016).

## 3.3 Compound hot-dry and associated impacts

Our results show that soybean production in southern regions is particularly sensitive to the co-occurrence of high August/September maximum Temperature and low August/September soil moisture (Fig. A3). The co-occurrence of low soil-moisture (5th percentile) and high temperature conditions (95th percentile) triggers the largest crop failures estimated at -2 standard deviations (calculated using spatially averaged model coefficients for August temperature, soil moisture and the interaction term). Extreme August hot-dry conditions (i.e. simultaneously exceeding the 95th and 5th percentiles of temperature and soil moisture, respectively) leads to 4 times more crop yield impacts compared to extreme hot conditions alone (i.e. 95th and 50th percentiles of temperature and soil moisture, respectively) and 3 times more impacts compared to extreme dry conditions alone (i.e. 50th and 5th percentiles of temperature and soil moisture, respectively). These results are qualitatively similar when we replaced August with September soil moisture. To further illustrate the implication of including interaction terms, we focus on Illinois in what follows. Illinois is the largest soybean producing region in the US and includes a large ratio of counties where summer moisture and temperature interactions are included in locally specified models (Fig.

A3b). Figure 5a shows pooled yield observations for Illinois (points) together with model predictions (contour lines) for various values of August root zone soil moisture (vertical-axis) and August maximum temperature (horizontal-axis). Qualitatively similar results are obtained when we replaced August with September root zone soil moisture. The coefficients for the

sensitivity of soybean yields to August hot-dry conditions in Fig. 5 are obtained from averaging all regression coefficients (i.e. for August Temperature, soil moisture and the interaction term) from all county-specific models within Illinois (i.e. 51 individual models/counties).

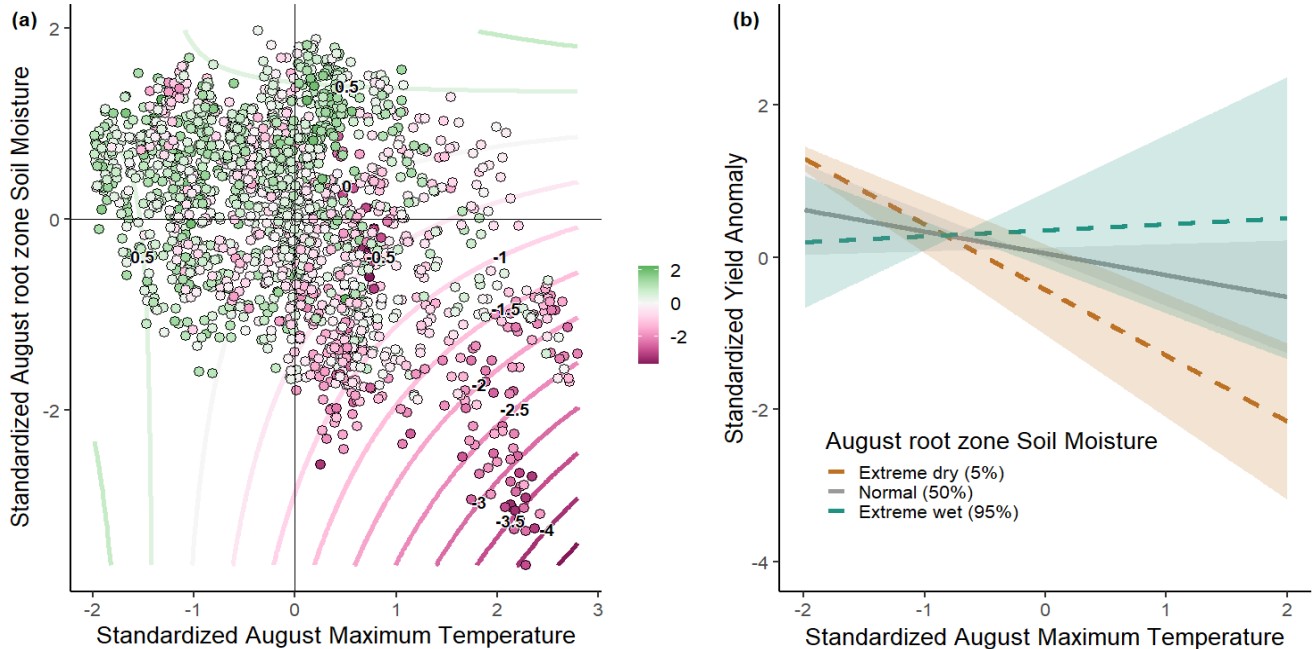

**Figure 5. (a) contour lines for modelled yield anomalies under varying levels of standardized August maximum temperature and root zone soil moisture in Illinois state. Points represent observed yield values. The colour scale to the right is in the units of standardized yield anomaly. (b) Sensitivity of Illinois US yield anomaly to temperature change for three different root zone soil moisture percentiles (5th, 50th, 95th ).**

Yield is shown to decrease for increasing hot-dry conditions both in observations and model predictions. In particular, the

bottom-right corner (representing August temperature and soil moisture values respectively above and below the 50th percentile) contains 75% of all observed low yields (defined as below one standard deviation). By including interaction terms, LOOCV-1 model performance improved by 17% for Illinois. In particular, we estimate that the compounding impact of hot-dry conditions (i.e. 95th and 5th percentiles of temperature and soil moisture, respectively) in August leads to an additional crop-loss of 0.6 standard deviations as compared to a model that includes all selected predictors but no interaction terms. On

the other hand, the effects of extreme hot-wet conditions (95th percentile for both temperature and soil moisture values) leads to a 0.5 standard deviation positive increase in crop yield estimates when including interaction terms. This non-linearity is visualized in Fig. 5b showing model-derived yield sensitivities to temperature for different levels of root zone soil moisture (i.e. 5th, 50th and 95th percentiles). The association between yield and August maximum temperature is strongly negative for

extremely dry conditions (brown dashed line) and slightly positive for extremely wet conditions (blue dashed line). This

highlights the importance of accounting for interaction effects when estimating compound impacts on crops. Yield response to hot-wet conditions is nevertheless subject to high uncertainty (see shaded uncertainty range in Fig. 5b) as these conditions do not occur often and are represented by few observations (upper-right corner in Fig. 5a). The rarity of these events is expected owning to the negative correlation between moisture and temperature over summer (Zscheischler and Seneviratne, 2017). It follows that wet conditions generally limit exposure rather than sensitivity to very high temperature. Still, temperature

sensitivities during wet conditions are significantly different from those during dry conditions (Fig. 5b).

### 3.4 Changes in compound hot-dry from 1946 to 2016

Linear trends for summer precipitation (JJAS) over the period 1946 to 2016 show significant increases particularly over the Midwest region (Fig. 6b). Only south-eastern states show significant drying trends. Maximum August temperature trends show significant cooling over the Midwest region but warming for north-eastern, north-western and southern states (Fig. 6a).

Moreover, early and late season minimum temperature trends indicate warmer conditions across the spatial domain whereas early season number of wet days trend indicates wetter conditions in spring (see Fig. A5). Though summers generally got wetter and cooler in the eastern part of the Midwest and north eastern US regions, the percent-change in the number of concurrent hot and dry summer months (i.e. 90th and 10th percentiles of August maximum temperature and summer precipitation, respectively) between 1946-1980 and 1982-2016 shows an increase in frequency here (Fig. 6c). This might have

implications as compound hot-dry events appear to have increased in frequency in high producing regions, despite the apparent cooling and wetting patterns identified by univariate trends.

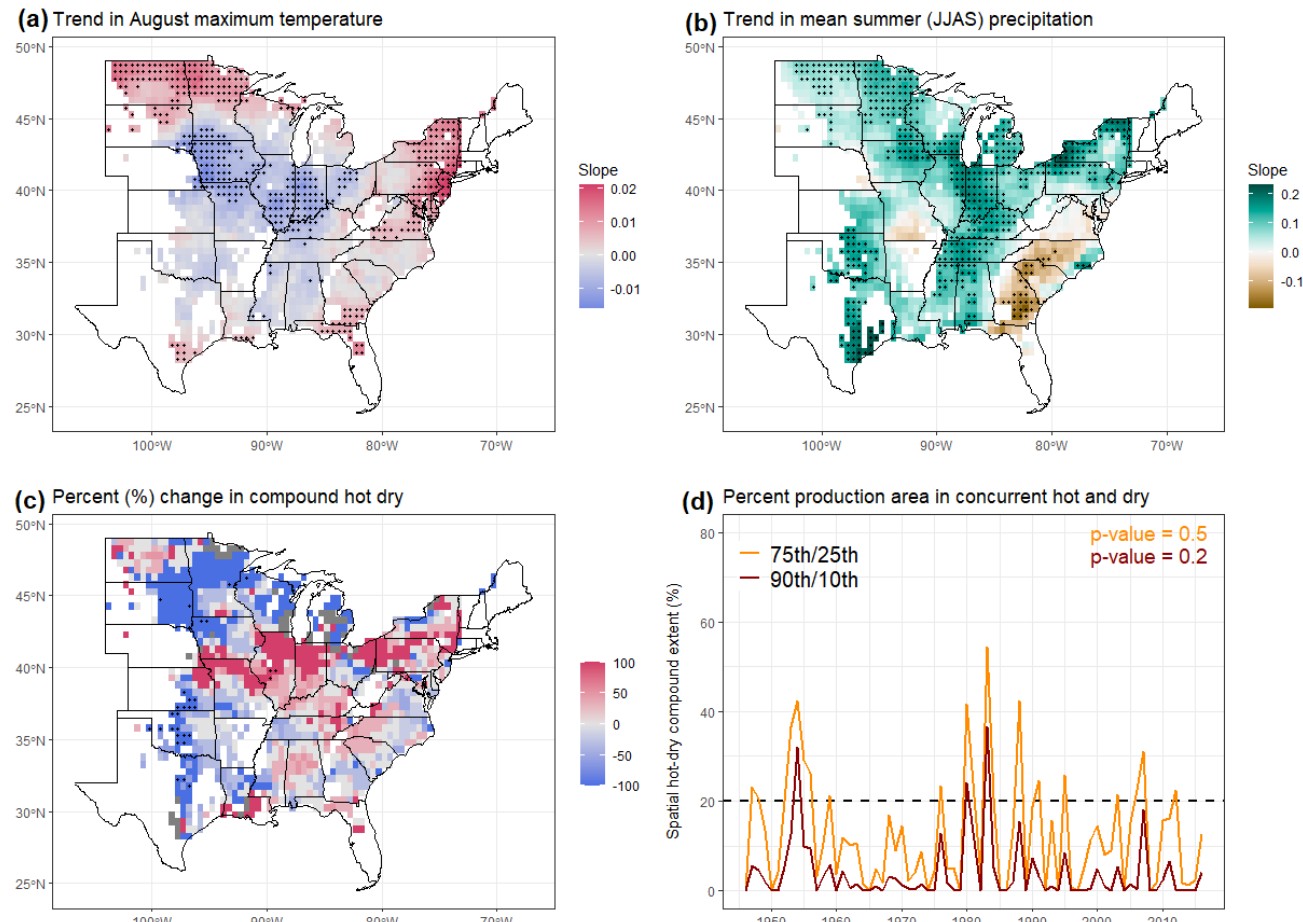

**Figure 6. (a) Linear regression slope of August maximum temperature. (b) Linear regression slope for summer (JJAS) precipitation. (c) Percent (%) change in concurrent dry (Summer JJAS precipitation < 10th percentile) and hot (August Maximum Temperature > 90th percentile) during 1982–2016 relative to 1946–1980. (d) Time-series of percent producing regions in hot and dry conditions. Trends in (a, b and d) are calculated for the period 1946 to 2016. Stippling in (a), (b) and (c) indicates statistical significance at the 95% confidence level. P-value in d) corresponds to the Mann–Kendall monotonic trend test. Black dashed line in (d) represents a 15% threshold marking years with a large (>15%) spatial hot-dry extent.**

Time series of percent production area in concurrent hot and dry conditions reflects the spatial extent of such conditions over the years (Fig. 6d). The black dashed line represents a threshold set at 20% exceeded by a number of years (i.e. 1947, 1948, 1953, 1954, 1955, 1956, 1959, 1976, 1980, 1983, 1984, 1988, 1991, 1995, 2003, 2006, 2007, 2012 when using the 75th/25th percentile hot-dry time series). More than 60% of those years coincide with La Niña like conditions, which have been shown to impact US crop production (Anderson et al., 2019; Iizumi and Sakai, 2020). Moreover, we note a high frequency of large-scale hot-dry events in specific periods such as the 1950s and 1980s. These segmented periods of high intensity events suggest a potential important role of decadal climate variability in the occurrence of hot-dry conditions. These can be related to low-frequency sea surface temperature variations such as the pacific decadal oscillation (PDO) shown to have an influence on local precipitation and temperature levels over eastern US (Vijverberg et al., 2020). A large fraction of the production area under

hot-dry conditions create risks for country level agricultural production as regions are no longer able to balance out losses at the local scale. Here again, despite the dominant cooling and wetting trends over the US (Fig. 6a & d), no significant up- or downward trend was found in the fraction of US under hot-dry conditions over time for both the 75th/25th and 90th/10th time-series.

### 3.5 Temperature and moisture couplings during summer hot-dry events

To better understand why compound hot-dry conditions have not changed, despite significant trends towards wetter summers and cooler August maximum temperatures, we analyse local land-atmosphere couplings. It has been hypothesized that during dry conditions, the actual evapotranspiration reduces, cancelling the land-change induced cooling effect and prompting a return to historic high temperature extremes (Mueller et al., 2016). Interannual correlation between root-zone soil-moisture (SMroot), maximum temperature (Tmax) and actual evapotranspiration (AET) pairs for a given month of the year, repeated over the various calendar months are used to estimate the coupling strength during hot-dry summer years and normal summer years. The subset of hot-dry events in this case is constructed from years when more than 20% of the total harvested area is under hot-dry conditions, defined using the 75th and 25th percentiles of August maximum temperature and summer precipitation (JJAS) respectively (i.e. years when the orange line is above the dashed black line in Fig. 6d).

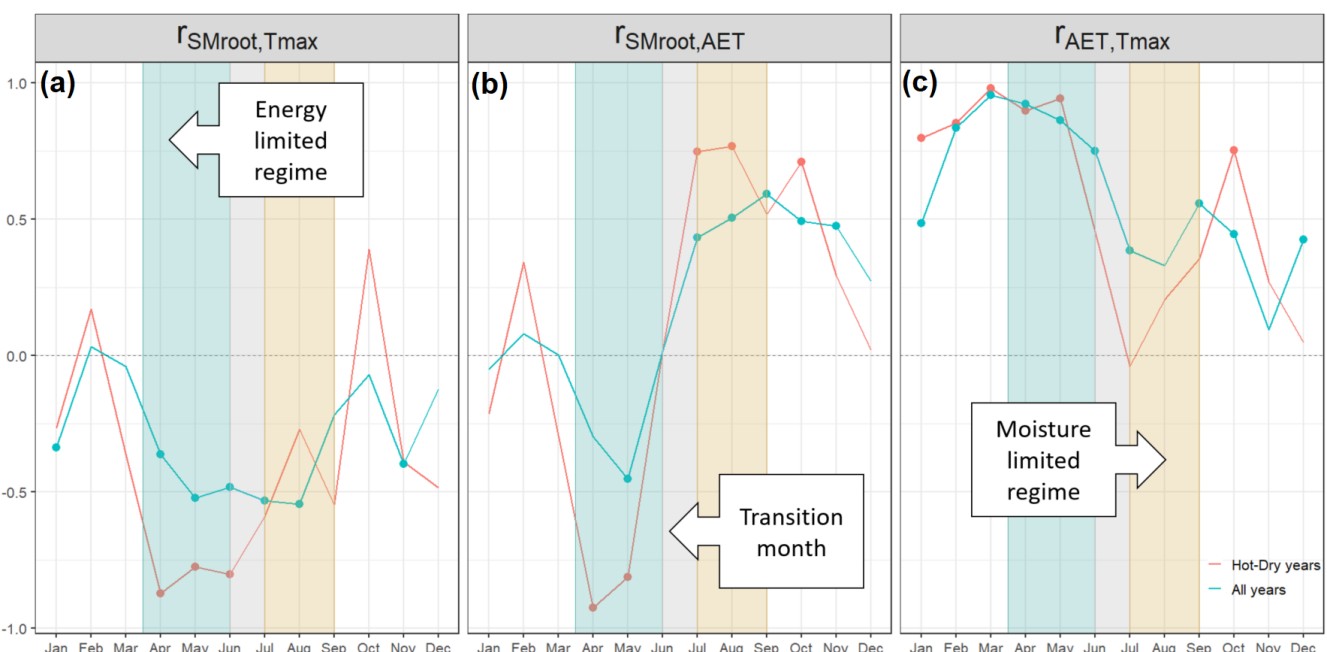

**Figure 7. Interannual correlation between various pairs of temperature and moisture variables for a given month of the year, repeated over the various calendar months conditioned on hot-dry events. Dots indicate statistical significance at the 95% confidence. Shaded regions represent important differences in the couplings that can play a critical role in the development of hot-dry events.**

We observe that summer hot-dry years are characterized by a stronger negative coupling between soil moisture and temperature during spring (April-May) when compared to a typical year (Fig. 7a). We interpret this negative coupling as indicative of

warmer and drier springs. These conditions create a stronger negative coupling between evapotranspiration and soil moisture as evapotranspiration rates are enhanced by warmer temperatures, in turn, rapidly depleting soil moisture reserves (Fig. 7b). The timing when the coupling between evapotranspiration and soil moisture sign shifts reflects a critical moment in the system when soil moisture becomes limiting. We observe that this regime-shift is much more pronounced during hot-dry years (i.e. stronger negative coupling in April-May and stronger positive coupling in July-August) (Fig. 7b). June is a transition month. The moment of the regime shift (around June) coincides with the ceasing of the spring coupling between evapotranspiration and temperature during hot-dry years (Fig. 7c). We interpret this ceasing of the coupling between evapotranspiration and maximum temperature as an indicator of total depletion of moisture in the soils, and thus extra energy (via higher temperatures) cannot lead to more evaporation. We are thus in a moisture-limited land-atmosphere coupling regime. During normal years, still significant coupling between evapotranspiration and maximum temperature exists in July-September indicating that the soils are not fully depleted. Spatially, the ceasing of the land-surface induced cooling effect is present over most of the soybean harvesting region going from June to September for hot-dry years (Fig. A6). To summarize, we show that summer hot-dry events are associated with warmer and drier springs. These conditions favour fast and intense depletion of soil moisture. Dry soils limit the evaporative cooling effect as captured by the annulled co-variability between actual evapotranspiration and temperature leading to amplified hot and dry conditions in summer (Fig. 7c). This provides evidence in support of the initial hypothesis that highlights the important role of land-atmosphere feedbacks in explaining the absence of a trend in summer hot-dry events despite summer wetting and cooling trends over the soybean production region in the US.

## 4 Discussion

Predictors here are determined statistically, nevertheless, we aimed for a restricted set of moisture and temperature variables for all US counties to facilitate the physical interpretation of climatic drivers affecting soybean yield variability. This is in line with other studies that constructed semi-empirical crop models relying on a statistical framework driven by well-known physiological variables (Ben-Ari et al., 2018; Gornott and Wechsung, 2016; Schauberger et al., 2017b). The frugal approach we used to select predictors implies that potentially useful and physiologically-relevant variables such as radiation and vapour pressure deficit are omitted. Although their effects can be implicitly accounted for in the temperature and moisture variables used, light exposure, for instance, certainly plays a key role in crop productivity (Farquhar et al., 2001; Rigden et al., 2020). Nevertheless, the choice is made as the least-squares model fit is highly sensitive to the ratio of predictors to the number of observations (James et al., 2013). Ideally, crop-observations (35 here) should be much larger than the number of predictors to avoid the risk of overfitting. Furthermore, including highly-correlated predictor variables (e.g. radiation and temperature) affect model parameter estimation and complicates physical interpretation of drivers. Future studies can disentangle these mechanisms for a more detailed data-driven assessment of climate and crop yield sensitivities. It is also possible to use more complex machine learning models such as random forests although these often tend to obscure result interpretation and do not always lead to better predictions (Vogel et al., 2019, 2021). Note that non-climatic seasonal influences on crop yields are

ignored in this study. These include planting densities, sowing dates, fertilizer applications and other socio-economic factors. This simplification is done as spatially-explicit time series for such components are rare and difficult to obtain (Schauberger et al., 2017b). Some of these factors were shown not to necessarily improve model performance in a case study done on crop yields in Germany (Gornott and Wechsung, 2016). Nevertheless, future studies should include these in whenever this becomes
possible for extended time periods as climate has been shown to influence seasonal management practices for farmers in the US (Carter et al., 2018b).

We found that soybean yields were predominantly driven by heat and drought conditions occurring during the vulnerable summer crop reproductive stage. In particular, August and September months were highlighted as key months for soybean
production in line with results from previous studies (Mourtzinis et al., 2015; Ortiz-Bobea et al., 2019; Zipper et al., 2016). Furthermore, we noted a significant interaction effect between summer maximum temperature and soil moisture variables modulating the final impact on yield. Drought and heat induce different growth inhibition patterns that can act simultaneously to reduce crop photosynthetic rates and eventual yield levels (Suzuki et al., 2014). August mean maximum temperature was found to be negatively associated with soybean yields for values exceeding 30°C (i.e. average August maximum temperature
value for a large part of the considered counties). This is in line with other studies that reported non-linear association between soybean and temperature where the relationship is mildly positive up until 30°C and then declines sharply due to heat stress (Schauberger et al., 2017a; Schlenker and Roberts, 2009). Moreover, here we found that this relationship was dependent on concurrent soil moisture conditions where wet soils dampen the negative effect of high temperatures on yield via evaporative cooling. A result that is also supported by previous studies reporting the decoupling effect of irrigation on the relationship
between heat stress and yield (Carter et al., 2016; Schauberger et al., 2017a; Schlenker and Roberts, 2009; Siebert et al., 2017; Troy et al., 2015). On the other hand, low moisture levels induce stomatal closure which leads to reduced latent heat flux and an increase in canopy temperature well above atmospheric temperatures increasing the crop sensitivity to hot conditions (Carter et al., 2016; Siebert et al., 2017). Such dependency highlights the important need to account for both variables simultaneously when assessing their impacts on crop yield variability (Carter et al., 2018a; Leng et al., 2016; Siebert et al., 2017; Suzuki et
al., 2014). Our analysis further reported early season excessive precipitation and minimum and maximum temperature conditions in addition to late season minimum temperature as important drivers of soybean yield variability. Early season excessive precipitation sensitivity likely reflects damaging plant field establishment conditions related to restricted root development, nutrient leaching and disease susceptibility (Li et al., 2019; Ortiz-Bobea et al., 2019). High minimum and maximum temperature in the early season positively associated to yield can imply both a reduced frost risk in addition to a
potentially longer growing season where soybean yield potential is maximized (Bastidas et al., 2008; Mourtzinis et al., 2019). End of season frost has also been reported as an important risk factor for soybean crops particularly in the northern states, and we interpret the predictor of minimum temperature during September and October as reflective of such conditions. These identified drivers of impact can serve as a basis for effective early warning systems that provide valuable information to decision makers (Merz et al., 2020). Acting in advance can be critical to avoid crop loss and associated socio-economic

consequences. For instance, a short period of drought during the reproductive stage is reported to cause non-reversible damage to soybean yields (Daryanto et al., 2017). Hot and dry conditions in eastern US over summer has been shown to be forecastable at long lead times (~50 days ahead), associated with sea surface temperature anomalies over the northern Pacific Ocean (McKinnon et al., 2016; Vijverberg et al., 2020). Future work can further explore the link between drivers of compound hazards impacting yields to facilitate the development of actionable tools for stakeholders.

We showed that historic changes in climate have not increased the overall climate risk for rainfed soybean production in the US. This is in line with other studies that looked at the contribution of historic climate trends on soybean and maize yields in the US (Butler et al., 2018; Ray et al., 2019). This is particularly the case in the most northern states where the occurrence of compound hot-dry events has mostly decreased (Fig. 6d). These regions are characterized by a predominantly energy-limited summer regime where the role of soil moisture in related land-atmosphere feedbacks is limited (Seneviratne et al., 2010). These northern states also showed reduced sensitivity to high temperatures over summer (Fig. 4d) in line with Lesk et al., (2021) who highlighted reduced soybean yield sensitivity to temperature in energy limited regimes at the global scale. Interestingly, soybean cropping regions have also shifted north-westerly in the US taking advantage of such changes in climate (Sloat et al., 2020). Increasing trend in number of wet days during spring can lead to detrimental change for rainfed soybean production. Nevertheless, Lesk et al., (2020) recently highlighted that the association between heavy rainfall and US crop yields can be different and more complex when studied at sub-daily resolution emphasizing that further investigation is needed in that regards. The summertime cooling is a well-documented phenomenon over US agricultural regions and is attributable to agricultural intensification in the region although other driving processes such as decadal variability and aerosol emissions also play a role (Alter et al., 2018; Lesk and Anderson, 2021; Mueller et al., 2016; Nikiel and Eltahir, 2019). With respect to the role of agriculture, a higher density of crops supported by increasing fertilizer rates leads to higher evapotranspiration rates which in turn induce large scale evaporative cooling and contribute to increasing precipitation (Basso et al., 2021; Mueller et al., 2016). Nevertheless, we highlighted that in key producing regions like Illinois, compound hot-dry events seem to have increased in frequency recently, despite the absence of a summer-mean drying or warming trend. Potentially, during dry conditions, the actual evapotranspiration reduces, cancelling the land-change induced cooling effect and prompting a return to historic high temperature extremes (Mueller et al., 2016). We illustrated this mechanism by analysing the evolution of land-atmosphere coupling within the growing season, captured by interannual correlations between root-zone soil-moisture (SMroot), maximum temperature (Tmax) and actual evapotranspiration (AET) pairs for a given month of the year, repeated over the various calendar months. We interpreted positive correlation values between actual evapotranspiration and maximum temperature as indicative of a general land-surface induced cooling effect. During hot-dry years, this evaporative cooling ceased at the onset of summer months. We showed that this was associated to stronger negative coupling between evapotranspiration-soil moisture and soil moisture-temperature in spring. Such conditions lead to fast soil moisture depletion and favour a moisture limited regime that amplifies extreme summer hot-dry conditions and associated soybean impacts (Sippel et al., 2016). Although we showed that warmer and drier springs lead to higher yields, potentially connected hot and

dry summer conditions lead to disproportionately negative impacts on final crop yields. Future risk assessments should account for such non-linear effects. Over the Midwest US, climate models project warmer summers which is likely to enhance the coupling between moisture and temperature via land-atmosphere feedbacks leading to a possible increase in the amplitude and frequency of compound hot-dry conditions (Cheng et al., 2019; Zscheischler and Seneviratne, 2017). Although annual precipitation levels are expected to remain constant or even increase, climate models generally project increased dry day length and decreased summer soil moisture levels (Dai, 2013; Dirmeyer et al., 2013; Wuebbles et al., 2014a, 2014b). Future research should quantify whether such trends could lead to an increase of hot-dry summer months in the future. Nevertheless, high uncertainty remains with respect to atmospheric dynamical changes including quasi-stationary Rossby waves which are a key driver of hot-dry conditions in the eastern US as well as other mid-latitude regions (Di Capua et al., 2020; Coumou et al., 2014; Kornhuber et al., 2019; Shepherd, 2014; Winter et al., 2015). Until such contradictions are resolved, future impacts of climate change on US agricultural production remain uncertain. The storyline approach has been proposed as an important tool to illustrate such epistemic uncertainty and can be explored in future studies with important consequences on current and future policy and decision making (Shepherd, 2019).

Here we focused on local types of compound events, however, global food supply is highly dependent on production in various countries. Spatially compounding events will be important to study in future assessments in order to understand large scale risk associated to breadbasket failures. Here we qualitatively identified that a considerable number of the large extent hot-dry conditions occurring over the US are coinciding with La Niña like conditions. These are also highly influential over the South American continent where soybean production including the US account for more than 80% of total global supply (Anderson et al., 2017; Iizumi and Sakai, 2020; Wellesley et al., 2017). Other examples of teleconnections are mid-latitude Rossby waves, particularly wave number 5, which has phase-locking behaviour in the northern hemisphere mid-latitudes driving simultaneous summer positive temperature anomalies over Midwest US, eastern Europe, and east Asia (Kornhuber et al., 2019). This is particularly of concern to soybean production when taking into consideration upcoming soybean hotspot production regions such as Russia and Ukraine (Deppermann et al., 2018).

## 5 Conclusion

We presented a simple statistical framework that can identify climatic variables influencing soybean yield variability in the US at specific moments within the growing season. We found that compound summer hot-dry conditions lead to the largest impacts on yield, i.e. beyond the estimated additive effects of each stressor separately. Furthermore, we identified early-season minimum and maximum temperature in addition to precipitation, and late-season minimum temperature and soil moisture to be important factors affecting soybean yield in the US. Understanding of these seasonally dependent crop-sensitivities paves the way for more effective early-warning tools that target timely drivers of yield variability throughout the growing season. The long-term cooling and wetting trend in summer, over large areas of our domain, has generally been beneficial for soybean.

Nevertheless, we showed that the frequency of extreme hot-dry conditions remained largely unchanged over the full region, and increased in a key region like Illinois where crops are especially sensitive to such extremes. Furthermore, we showed that hot-dry events are characterised by stronger negative spring coupling between evapotranspiration-soil moisture and soil moisture-temperature leading to fast soil moisture depletion in spring and a reversal in the land-surface cooling mechanism over summer prompting important soybean yield impacts. Given that climate models project summer warming and general declines in soil-moisture (albeit with substantial uncertainty) for the Midwest, crop sensitivities to compound hot-dry extremes are likely to present important future risks for US soybean production.

**Appendix A: Additional figures**

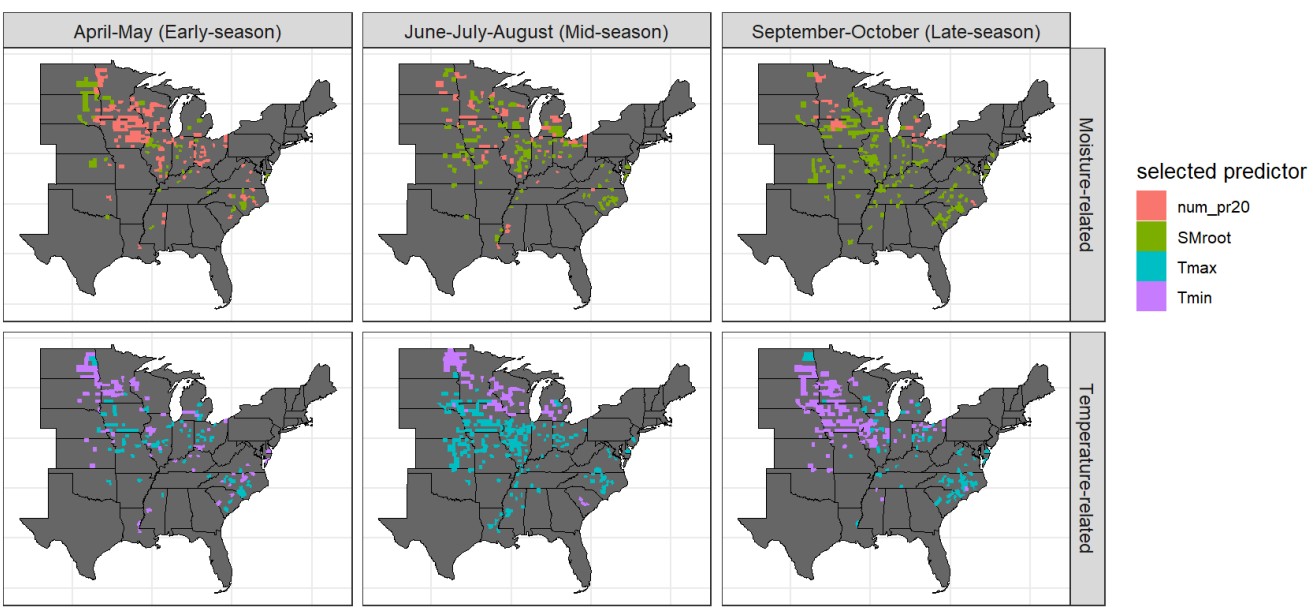

Figure A 1. Selected predictors per county based on the full dataset

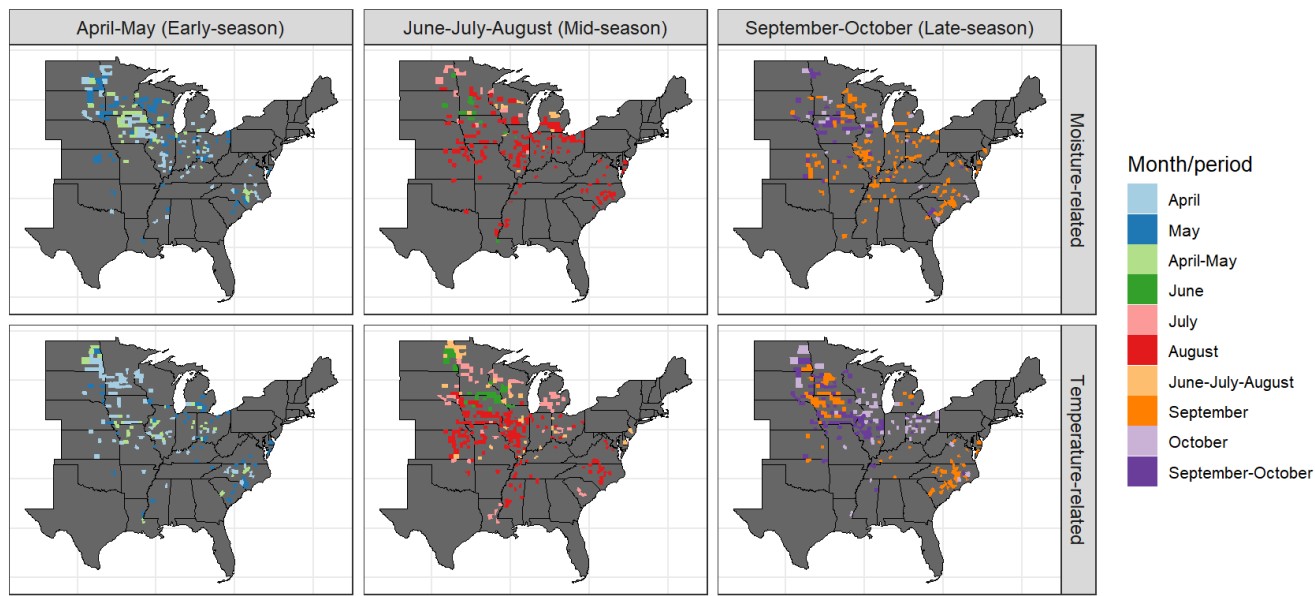

**Figure A 2. Selected timing of predictors per county based on the full dataset**

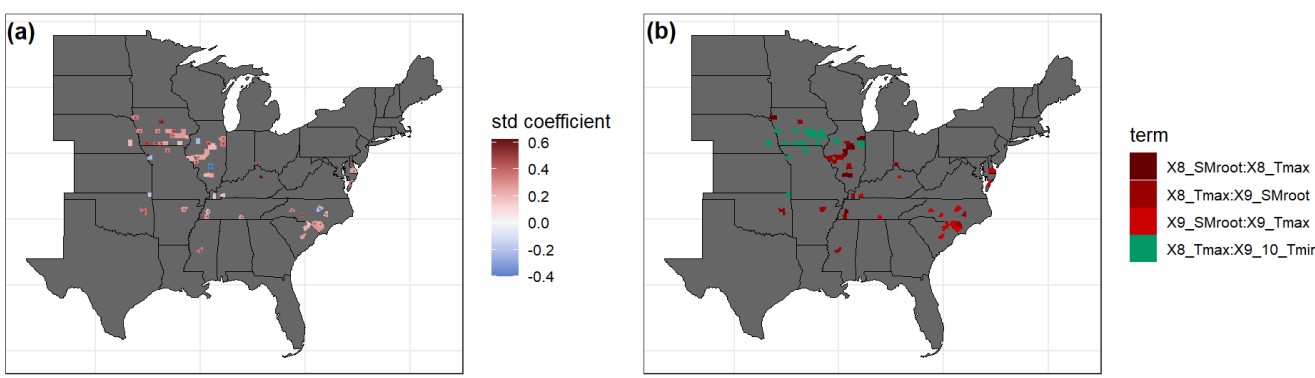

**Figure A 3. (a). Standardized coefficients for interaction terms per county selected based on the full dataset. (b). Type of interactions selected per county based on the full dataset.**

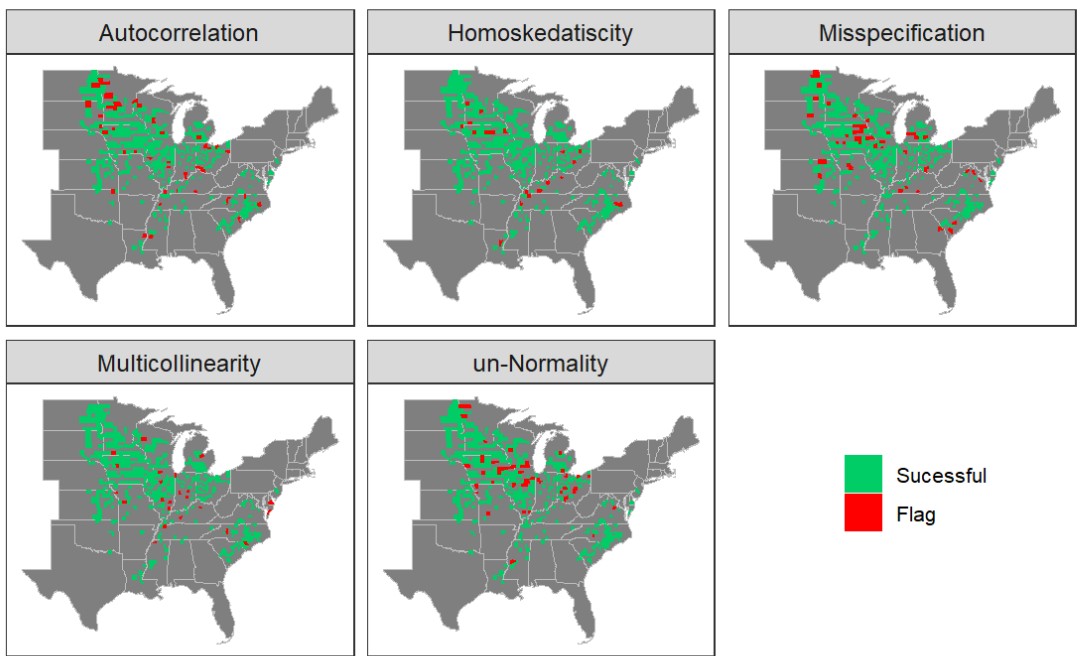

**Figure A 4. Diagnostic test results for the fitted models. Green indicates a "successful" test, i.e. no problem, while red indicates a rejection of the respective H0 of no autocorrelation/heteroscedasticity/ misspecification/multicollinearity/un-normality. Multicollinearity is checked with the variance inflation factor and marked in red if any of the variables report a value >3.**

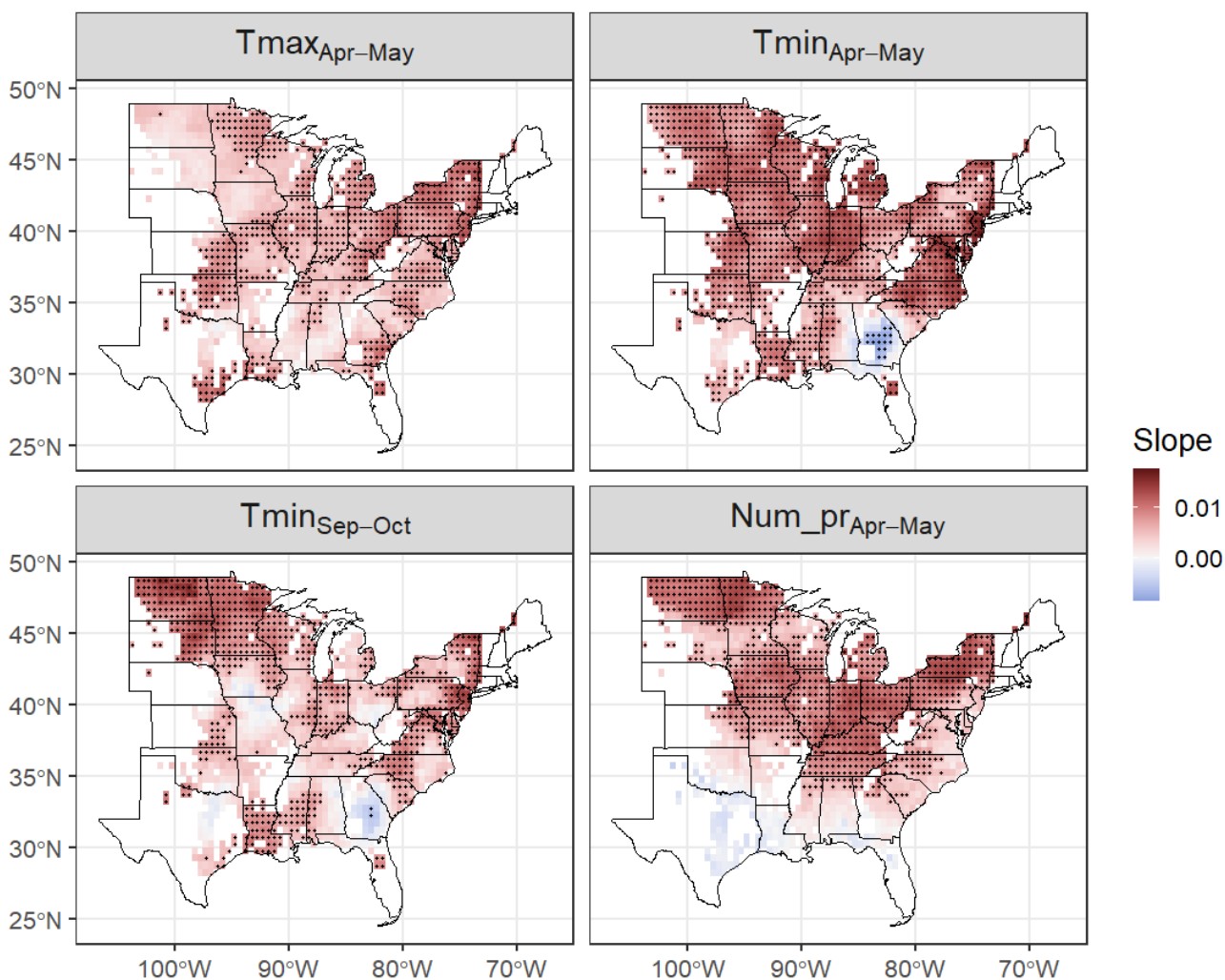

**Figure A 5. Linear trends for main identified drivers of soybean yield variability over the period 1946-2016. Stippling indicates statistical significance at the 95% confidence level. Trends for moisture and temperature variables over summer are displayed in the main text.**

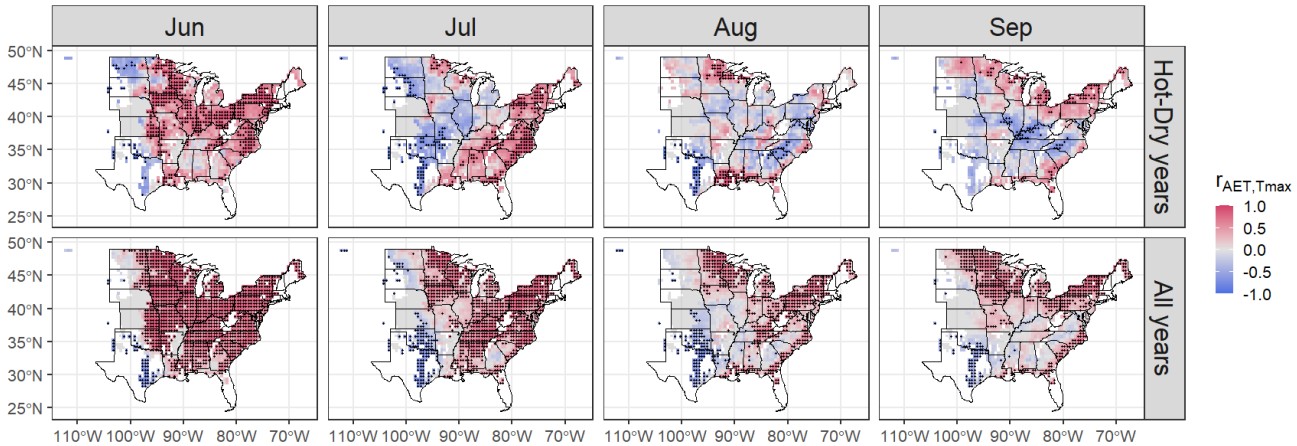

**Figure A 6. Interannual correlation between actual evapotranspiration and maximum temperature for a given month of the year conditioned on hot-dry events and repeated for the period going from June to September. Dots indicate statistical significance at the 95% confidence.**

*Code availability*. The code is available upon request, by contacting the corresponding author.

*Data availability*. Data used in this study are freely available in the cited literature.

*Author contributions.* RH and DC designed the study. RH performed the analysis and wrote the initial draft of the paper. All authors contributed to the development of the analysis, the interpretation of the results and to the writing of the paper.

*Competing interests.* The authors declare that they have no conflict of interest.

*Special issue statement.* This article is submitted to the special issue "Understanding compound weather and climate events and related impacts".

*Acknowledgements*. We thank the various institutes cited in the text for making the data used in this study freely available. We also would like to thank the reviewers for helping us improve the content of this paper and for their encouraging words.

*Financial support*. This research has been supported by the European Union's Horizon 2020 research and innovation programme under grant agreement No 820712 (project RECEIPT, REmote Climate Effects and their Impact on European sustainability, Policy and Trade).

*Review Statement*. This paper was edited by Jakob Zscheischler and reviewed by Corey Lesk and one anonymous referee.

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
