# Peer review of "Impacts of hot-dry compound extremes on US soybean yields"

_Earth System Dynamics, 2021_

## Author Response (AR1)

We would like to thank the reviewers for their helpful and constructive comments, which, we believe, considerably helped improving the quality of the submitted manuscript. Please find below a list of all relevant changes made in the manuscript followed by a point by point response to the reviewers (responses in italic and bold). Referenced Line numbers refer to the track changed version of the manuscript. The point by point section is almost similar to the previously submitted responses to reviewers on the manuscript portal. Differences relate to omitting most figures and referring to them as they are currently presented in the revised manuscript and supplementary material. Furthermore, for the additional analysis requested by the first reviewer, we now compare a subset of hot-dry years to all considered years instead of remaining years as initially presented in the reviewer. Results remain qualitatively similar.

**List of all relevant changes:**

- 1- The initial list of potential predictors is reduced to Minimum Temperature, Maximum Temperature, Root Zone Soil Moisture and Excessive precipitation. Table 1 is updated accordingly to include these changes.
- 2- The manual variable exclusion in the predictor selection step is omitted.
- 3- The predictor selection and model fitting is now set to run strictly at county scale.
- 4- A robust one out of sample cross validation including the predictor selection step is added to the analysis. Supporting figures (Fig. S3, S4 and S5) with respect to the robustness of selected predictors are added to the supplementary material.
- 5- Figure 2 has been updated to include the latest changes with respect to the statistical modeling framework
- 6- Figure 3 has been updated to reflect changes with respect to the statistical modeling framework. An additional panel has been included to summarize selected predictors and strength of calculated beta coefficients over the study region.
- 7- Figure 4 has been updated and modified to reflect changes with respect to the statistical modeling framework. As predictors are now selected at county scale, Fig. 4 displays coefficients for the various selected moisture and temperature variables per county across the different stages of the growing season (i.e. early-, mid and late season). Details on the type of selected predictor and timing within the season are added to the Appendix (Fig. A1 and A2).
- 8- Figure 5 in the main text is updated to include results from the latest model setup.
- 9- Summer precipitation conditions previously calculated as averaged value over JJA months now further includes Sep month (i.e. JJAS) to reflect relevance of soil moisture in Sep (Fig 3a).
- 10- Figure 6- d in the main text is updated to also include hot-dry events defined according to the 75th August maximum temperature and 25th summer moisture percentiles respectively.
- 11- To better understand why compound hot-dry conditions have not changed, despite significant trends towards wetter summers and cooler August maximum temperatures, we add further analysis in the manuscript to explore the potential role of local land-atmosphere couplings in the outset of such hot-dry conditions (Sect. 3.5). Monthly correlations between root-zone soil-moisture (SMroot), maximum temperature (Tmax) and actual evapotranspiration (AET) are used to estimate the coupling strength during hot-dry summers and normal summers. The subset of hot-dry events in this case is constructed from years when more than 20% of the total harvested area is under hot-dry conditions, defined using the 75th and 25th percentiles of August maximum temperature and summer precipitation (JJAS) respectively. Figure 7 is added to the manuscript to summarize those results.
- 12- Figure A3 is added to the appendix showing selected model interactions via the stepwise approach.
- 13- Figure A5 has been updated to focus on main predictors as determined from the updated statistical modeling framework. Primarily, this concerns the inclusion of excessive precipitation in the early season as predictor. Furthermore, we've considered number of wet days from the CRU dataset to be able to explore trends in excessive precipitation over a longer period. Correspondence between the CRU time series and the originally considered variable for excessive precipitation from the W5E5 dataset is determined via correlation analysis. Resultant figure is added to the supplementary material Fig. S2.

**Point-by-point response to the reviewers:**

**Referee 1: Corey Lesk**

This paper examines the dependence of county level historical soybean yields on a suite of climate variables, with a particular eye to the influence of compound extremes. The paper is well motivated by recent climate and crop science literature, and nicely illustrates the particular relevance of hot and dry extremes for soybean yields. It further outlines historical trends in such extremes in relation to their univariate temperature and soil moisture components (using proxies to extend the time series). The manuscript is well written and the discussion raises a lot of interesting points. Overall, I think the authors did a great job and the paper should be considered for publication with revisions.

To me, the weaknesses of the paper are a 1) that there are a few methodological concerns and 2) that the paper doesn't extend that much beyond what is already fairly well established, even though it could using its data and methods. I elaborate below on these critiques and suggest some ways the authors could make the paper more compelling in revision.

**RESPONSE:** We thank the reviewer for the positive feedback on our manuscript. We are grateful for the critiques and suggestions on how our manuscript can be improved. We respond to the comments given in the text below (in **bold** and **italics** text).**

General comments

**Methodological concerns:**

The statistical modeling framework is quite detailed and meticulous and there is much attention paid to many sources of confusion, error, or interpretability issues. I commend these solid methods. However, I think the contradiction in scale between the model calibration (pooled national data) and its application (county-level) data greatly limits the soundness of the otherwise meticulous method. This to me is one of the main things to address in revisions.

The use of pooled national data to calibrate the model and then the subsequent use of that model to assess yield variability at county scales seems a bit inconsistent. All county-year pairs of yield values were combined into a larger dataset as in the Troy et al. (2015) paper referenced, but then the resulting model was applied for individual counties, whereas Troy et al. ran all their analysis at national scale. It doesn't seem common or intuitive to me to calibrate and run models at such different scales, and this isn't justified, discussed, or acknowledged in any detail. What the authors did here is an expedient way to make their analysis applicable at a county level, but leverage a wider dataset to increase degrees of freedom and enable the testing of more complex models. But the cost is that the mismatch in scales raises questions on whether some results are a result of the mismatch, or truly robust results.

A few ideas on how this might be influencing results: First, it could be that the particular relevance of concurrent low August soil moisture and high Tmax in Illinois (nicely illustrated in Fig. 5) is a result more of the suitability of the nationally fitted model at that location, rather than the relevance of compound extremes more generally. This concurrent temperature-moisture result does not strongly agree with the results of Mourtzinis et al. (2015) which found larger temperature impacts in southern states, nor Zipper at al. (2016) which found stronger drought impacts on soy in southern states. It could very well be that the compound impact is larger in Illinois, but the ambiguity induced by the contradiction in scales in the modeling makes the result not as robust as it could be.

Second, there is strong variation in the significance and model r2 across counties (Fig. 3). Generally, this itself raises the question of whether models should be calibrated locally (indeed we wouldn't expect nationally-consistent model to be optimal everywhere, even if parameters of said model are estimated locally). More specifically, the model performs quite strongly in Illinois, which to me raises the question of whether the strong compound Tmax-SM impact is really just because the national model works very well in Illinois (i.e., a result determined by methods, so not very robust).

The data-driven approach also is attractive because it allows the 'most important' months to be identified. However, I have doubts about the methods for this around the selection of the earlier among collinear climate signals (see detailed comments). Further, it's not only collinearity in time, but among

variables at the same time, that matters, as you discuss in lines 280-90. Leaf- to field-scale experiments show that light is very important for crops, so that it is excluded from the modeling is a methodological choice that requires careful interpretation. Another example is the idea that temperature is a strong predictor because it encapsulates many moisture and heat related stressors, as in many of the cited papers. Point is, as a result of these assumptions regarding variable exclusion in the methods, the model specification is actually not as data-driven in the end, so worth considering other approaches with their own strengths/weaknesses:

I think an alternative approach could be to compromise a bit on data-driven model specification and simply prescribe the model structure a priori. This is appropriate because you cite much literature in your introduction on why and how compound extremes should matter, so you have a prior to base the specification on. You can then run the stepwise model selection on a smaller set of predictors for each county and see if results are robust, e.g. Illinois/August still pops out. I understand there is a compromise in this alternative, but it might complement and add confidence given concerns about the original approach. It might also add some confidence to run the panel regression for the full national model (i.e. what is the national value of the coefficients in Fig. 4a).

RESPONSE: We thank the reviewer for generally commending the statistical framework and agree with his main criticism with regards to the mismatch in scale between predictor selection at national scale and model fitting that took place at county scale. Our initial intent was to conclude one unique set of predictors for all counties to facilitate interpretability and avoid potential overfitting at the local level. Nevertheless, we agree that such approach means that predictors are not optimal for each county which can introduce ambiguity when interpreting results. Furthermore, we do agree that the selection of the earlier among collinear climate signals needs further justification. In order to address these issues, we re-ran our analysis with a modified methodology in line with the reviewer's suggestions (see Fig. 2). The predictor selection and model fitting is now set to run strictly at county scale.

In order to reduce overfitting concerns, we ran a model one out of sample cross-validation that includes a predictor selection step (Robust-OOS) and limited the number of potential predictors to: Minimum Temperature, Maximum Temperature, Root Zone Soil Moisture and Excessive precipitation. These predictors are supported by main findings in prior literature that highlight the damaging effects of chilling conditions, high temperature, water stress and excessive rainfall on crops grown in the US (Carter et al., 2018; Gu et al., 2008; Li et al., 2019; Mourtzinis et al., 2015, 2019; Ortiz-Bobea et al., 2019; Zipper et al., 2016). The choice to exclude actual evapotranspiration and shortwave radiation from the modeling step is motivated by our intent to particularly focus on the effects of temperature and moisture climate variables on soybean yields and facilitate interpretation of results. Shortwave radiation and actual evapotranspiration are important variables with respect to physiological mechanisms in crop growth. Nevertheless, shortwave radiation is often highly correlated with temperature in the summer and actual evapotranspiration is influenced by soil moisture, temperature and crop growth making it particularly tricky to study all these variables together in a simple regression framework. This is not to say that there is no benefit in including these variables alongside temperature and soil moisture in a data-driven framework but such exercise requires further careful analysis that is not the focus of this work. These decisions are emphasized in the main text. Line 448-459

With respect to the selection of the earlier among collinear climate, we no longer intervene manually with predictor selection and only monitor multicollinearity concerns using the variance inflation factor (VIF). In the latter case, a flag is raised if the VIF exceeds a value of 3 for any variable used to fit the final model at county scale (Carter et al., 2016; James et al., 2013). The resulting general model performance, county scale coefficients, selected predictors and associated timing within the season are represented in Fig. 3b, 4, A1 & A2 respectively. Further robustness plots with respect to selected predictors and associated timing are represented in Figures S3, S4 and S5. These plots display the most frequently selected predictors and associated timing in addition to how frequently they've been selected across the robust one out of sample cross validation (Robust-OOS).

**Finally, in the submitted preprint, Illinois was highlighted for being a major soybean producing state in addition to the relevance of summer moisture-temperature interaction terms within the state. Southern states still showed in initial results, and in updated ones, following revision, strong sensitivity to temperature and moisture generally in line with Mourtzinis et al. (2015) and Zipper at al. (2016) (see Fig. 4).**

Novelty and advancing understanding:

I really like how this paper clearly puts data and nice visualization to the idea and existence of examples of compound extreme impacts on crops. It also goes into some detail on soybean, a crop for which there is somewhat less attention on the topic. However, I think the core conclusions of this paper have essentially already been established. For instance, the Illinois case study in Figure 5 is an excellent visualization, but its message essentially quite similar to Kent et al. (2015, Fig. 2c, a great paper on maize which might be a handy reference to include) combined with what was published in Matiu et al. (2017), namely that such compound impacts do occur in places. It's useful that this paper points out that this occurs for this particular crop and location, but a similar point has been made in Ortiz-Bobea et al. (2019, also a great paper that probably needs to be referenced in this paper). Examining trends in concurrent heat-drought is also a useful topic, but has been covered in some detail in e.g. Sarhadi et al. (2019) and Lesk and Anderson (2021). Overlap with past research is a great contribution, but I think it does demand that this study go a bit deeper.

For instance, I think the study could choose one result to go into some more detail on to really gain some new insight. One could be how exactly these extremes are impactful in some places, less so in others, and some of the uncertainties and challenges around understanding this (see minor comments). If the particular importance of compound impacts in Illinois turns out to be a robust finding, why exactly might this be? The authors hypothesize a link to a reversal of the crop induced land-surface cooling during dry episodes, leading to compound impacts (as suggested by Mueller et al. 2016). Many papers have recently speculated about this, and you have the data to examine this in great detail for this location (e.g. compositing and examining coevolution of AET, SM and Tmax timeseries over hot-dry events) and add valuable concreteness to the speculation. Another direction could be to assess drivers of the trends in Figure 6 more concretely, possible roles of agriculture itself in influencing those trends, roles of modes of climate variability and aerosols (e.g. Fig. 6c probably shows some dependence of on the changepoint selection as visible in Fig. 6d, why that might be, and does it say anything meaningful about future change?).

References:

Kent, C., Pope, E., Thompson, V., Lewis, K., Scaife, A. A., & Dunstone, N. (2017). Using climate model simulations to assess the current climate risk to maize production. Environmental Research Letters, 12(5), 054012.

Lesk, C., & Anderson, W. (2021). Decadal variability modulates trends in concurrent heat and drought over global croplands. Environmental Research Letters, 16, 055024.

Ortiz-Bobea, A., Wang, H., Carrillo, C. M., & Ault, T. R. (2019). Unpacking the climatic drivers of US agricultural yields. Environmental Research Letters, 14(6), 064003.

Sarhadi, A., Ausín, M. C., Wiper, M. P., Touma, D., & Diffenbaugh, N. S. (2018). Multidimensional risk in a nonstationary climate: Joint probability of increasingly severe warm and dry conditions. Science advances, 4(11), eaau3487.

RESPONSE: We thank the reviewer for highlighting the visualization component and are grateful for suggestions to further expand the analysis to include more insight into our work. We are also thankful for the suggested papers that we will include in our reference list. Focusing on the hypothesized reversal of the crop induced land-surface cooling effect during hot-dry summer years leading to extreme impacts is a particularly insightful and relevant section to add to our work. In an attempt to illustrate this, we analyzed local land-atmosphere couplings. Monthly correlations between root-zone soil-moisture (SMroot), maximum temperature (Tmax)

and actual evapotranspiration (AET) are used to estimate the coupling strength during hot-dry summers and normal summers. The subset of hot-dry events in this case is constructed from years when more than 20% of the total harvested area is under hot-dry conditions, defined using the 75th and 25th percentiles of August maximum temperature and summer precipitation (JJAS) respectively. We observe that summer hot-dry years are characterized by a stronger negative coupling between soil moisture and temperature during spring (April-May) when compared to a typical year (Fig. 7a). We interpret this negative coupling as indicative of warmer and drier springs. These conditions create a stronger negative coupling between evapotranspiration and soil moisture as evapotranspiration rates are enhanced by warmer temperatures, in turn, rapidly depleting soil moisture reserves (Fig. 7b). The timing when the coupling between evapotranspiration and soil moisture sign shifts reflects a critical moment in the system when soil moisture becomes limiting. We observe that this regime-shift is much more pronounced during hot-dry years (i.e. stronger negative coupling in April-May and stronger positive coupling in July-August) (Fig. 7b). June is a transition month. The moment of the regime shift (around June) coincides with the ceasing of the spring coupling between evapotranspiration and temperature during hot-dry years (Fig. 7c). We interpret this ceasing of the coupling between evapotranspiration and maximum temperature as an indicator of total depletion of moisture in the soils, and thus extra energy (via higher temperatures) cannot lead to more evaporation. We are thus in a moisture-limited land-atmosphere coupling regime. During normal years, still significant coupling between evapotranspiration and maximum temperature exists in July-September indicating that the soils are not fully depleted. Spatially, the ceasing of the landsurface induced cooling effect is present over most of the soybean harvesting region going from June to September for hot-dry years (Fig. A6). To summarize, we show that summer hotdry events are associated with warmer and drier springs. These conditions favor fast and intense depletion of soil moisture. Dry soils limit the evaporative cooling effect as captured by the annulled co-variability between actual evapotranspiration and temperature leading to amplified hot and dry conditions in summer (Fig. 7c). This provides evidence in support of the initial hypothesis that highlights the important role of land-atmosphere feedbacks in explaining the absence of a trend in summer hot-dry events despite summer wetting and cooling trends over the soybean production region in the US.

With respect to other drivers of the observed trends, we referenced in the revised manuscript works that investigated decadal variability, role of aerosols and the role of agriculture itself on temperature and moisture trends in the region (Alter et al., 2018; Lesk and Anderson, 2021; Mueller et al., 2016; Nikiel and Eltahir, 2019). Line 517-518

Detailed comments

Lines 25-30: Introduction has great context for why we should care about US soybean. I think it would give helpful context to readers to stay somewhere here that a large portion of soybean is produced for feed.

Cassidy, E. S., West, P. C., Gerber, J. S., & Foley, J. A. (2013). Redefining agricultural yields: from tonnes to people nourished per hectare. Environmental Research Letters, 8(3), 034015.

**RESPONSE:** Thank you for highlighting this important element. We included this information in the revised manuscript. Line 30-31**

Line 46: Agreed that more attention is needed especially for soy, but probably should cite a few missing studies that have be written on the topic, consider: Rigden et al. (2020), Ortiz-Bobea et al. (2019), Haqiqi et al. (2021)

Haqiqi, I., Grogan, D. S., Hertel, T. W., & Schlenker, W. (2021). Quantifying the impacts of compound extremes on agriculture. Hydrology and Earth System Sciences, 25(2), 551-564.

Rigden, A. J., Mueller, N. D., Holbrook, N. M., Pillai, N., & Huybers, P. (2020). Combined influence of soil moisture and atmospheric evaporative demand is important for accurately predicting US maize yields. Nature Food, 1(2), 127-133.

**RESPONSE:** Thank you for the suggested references. We included these in the revised manuscript.**

Line 91: There's some evidence that 30mm/day is not a high enough rainfall amount for negative impacts on soy yields in the US (Lesk et al. 2020). I wonder if heat/extreme rainfall would pop out as a compound (possibly positive/compensating) impact on crops if you used a higher threshold.

Lesk, C., Coffel, E., & Horton, R. (2020). Net benefits to US soy and maize yields from intensifying hourly rainfall. Nature Climate Change, 10(9), 819-822.

RESPONSE: Thank you for the reference. When comparing the two predictors: Number of days with precipitation above 20mm to number of days with precipitation above 30mm, we saw more often the selection/relevance of the earlier for the fitted models. We therefore used Number of days with precipitation above 20mm to gauge the effects of excessive precipitation on yield in line with Zhu and Troy (2018). The highlighted paper (i.e. Lesk et al., 2020) differs from our analysis in that it uses hourly data vs a coarser time resolution considered in our study. In addition, it focuses on the full growing season and doesn't account for timing with respect to the growing season. In our analysis, we show that the negative effects of heavy precipitation are almost exclusively occurring in the early season (Fig. 3a). Physiologically, this likely reflects damaging plant field establishment conditions related to restricted root development, nutrient leaching and disease susceptibility (Li et al., 2019; Ortiz-Bobea et al., 2019). Nevertheless, we acknowledged the reviewers' point in the revised manuscript by adding to the text: "Nevertheless, Lesk et al., (2020) recently highlighted that the association between heavy rainfall and US crop yields can be different and more complex when studied at sub-daily resolution emphasizing that further investigation is needed in that regards. " Line 513-515.

Line 115: Selecting the earlier among collinear monthly predictors raises an interesting question of whether the signal for one variable preceding the others in time necessarily means that variable is the driver of the crop response. That is, the later signal could easily have caused the real impact on the crop, and the earlier one is predictive because of its correlation with the later. This is worth justifying more, or at least acknowledging as an important assumption (because it partly determines what variables ultimately can be considered drivers of compound impacts in your methodology). Could be an angle for going to deeper on why Illinois pops out for example.

**RESPONSE:** We acknowledge that the choice made to select the earlier among collinear variables was more motivated by a practical concern rather than a causal framing. In order to avoid arbitrarily omitting one of these two variables as this was also a concern raised by the second reviewer, we decided to exclude this manual selection step as discussed above and only monitor multicollinearity concerns with the VIF value.**

Lines 123-5: Would be good to see more detail on which/why other interactions were left out, and exactly how much 'better' the selected interaction was than other candidates, as this is key to your conclusion. An weaker alternative could be to simply assert that this interaction is one you have a good reason to care about (i.e. the hypothesized interaction is the motivation of your analysis).

RESPONSE: In the revised methodology, all pairs of interactions between selected predictors are considered at county level and only dropped later via the stepwise approach. Figure A3 represent the location and pair of picked up interactions in the final model specification. Most interactions are related to hot-dry summer conditions except for the positive interaction picked up between August maximum temperature and September-October minimum temperature mainly around the state of Iowa. Although we don't focus much on the latter, this might be related to increased impact whenever conditions go from anomalously hot in August to anomalously cold in September-October further stressing crops and reducing the potential

**positive effects of crop temperature acclimation (see (Butler and Huybers, 2013; Carter et al., 2016) and references therein).**

Line 148: Ref needed for energy limited AET

**RESPONSE:** This line is omitted from the revised manuscript as it is no longer needed. We now discuss energy limited and moisture limited regimes in the revised manuscript following results presented in Sect. 3.5.**

Lines 165-6: I'm surprised SM and Tmax are not more strongly collinear in August given the landatmosphere feedbacks and their involvement in the compound extreme. This should be discussed more and possibly examined in depth. E.g. – are the feedbacks really setting up earlier in the season, so SM and Tmax are more collinear then, and thus get excluded from the analysis? If so, this raises questions of whether August then really is the most important for yield, or just popping up because of this methodological decision (although some other papers you cite do support August being important). There's something deeper to understand here.

RESPONSE: In the adjusted methodological setup, no additional selection steps are imposed besides the initial univariate BIC selection step followed by the stepwise regression approach. With regards to the land-atmosphere feedbacks, we hope the additional analysis presented under sect. 3.5 somehow illustrates how soil moisture, actual evapotranspiration and temperature correlation evolves across the season leading up to summer damaging compound extremes. August still pops up as very important month for yields even after allowing predictor selection to run at county scale (see Fig. A2). Robustness with regards to the selected predictors is further investigated with a cross-validation step that includes predictor selection iteratively highlighting again the relevance of August temperature in predicting soybean yield variability (see Fig S4 and S5).

Lines 176-8: I don't see this result supported by data in Fig. 3A, please explain.

**RESPONSE:** Thanks for pointing this out. The reference was initially intended to Figure 4 where coefficients used to calculate those values are presented. We adjusted accordingly in the revised text.**

Line 185: interesting that model predicts yields better in south (as in Schauberger) – crops here not necessarily 'decoupled' from climate, as warmer seasons benefit yields...

**RESPONSE:** We agree with this point and this is now better represented in the manuscript by allowing predictors to be selected at the local scale. Northern states do show that a warmer season even during summer benefits yields. Text adjusted Line 264-265**

Fig 3b: The question this raises for me is if the north-south gradient in r2 relates to a gradient in suitability of the nationally tuned model. Indeed, since Illinois is a major soybean producer, it's contribution of data to the pooled sample is particularly high (meaning the strength of the prediction could be because the national model fits best there, while other models would fit just as well if calibrated on smaller scales).

**RESPONSE:** We agree with this point and have adapted the methodology accordingly as discussed earlier. In line with reviewer's concern, we indeed did find that the northern regions are actually more sensitive to cold conditions rather than hot conditions during the summer period and these are mostly around the earlier month of summer (i.e. June and July) (Fig. A1 and A2). As predictor selection is now executed at the county scale, selecting such predictors for the northern states did improve  $R^2$  for this area and reduced the north-south gradient in model performance (Fig. 3c).

Lines 192-3: how does this square with e.g. Li et al. 2019 who show both very high and very low soil moisture are damaging?

RESPONSE: In this revised model setup, both soil moisture and excessive precipitation are included as covariates of soybean yields with high levels of soil moisture predominantly positively influencing yields and excessive precipitation negatively affecting yields. This is to say that we do find evidence that both very low and excessive moisture are damaging for crops in line with Li et al. 2019. The fact that the RESEST test shows that most model fits would have not improved had we considered quadratic variables is possibly related to the following factors. First, the quadratic association between moisture and yields is more pronounced depending on the month considered. During August, it may be that most losses are related to drought conditions in line with (Li et al. 2019, Figure 1-d). Second, during initial exploratory data analysis we did, we noted that seasonal climatic averages compared to monthly averages showed much more clearly the quadratic relationship to yield. Finally, we also noted that soil moisture had a less pronounced quadratic relationship to yield when compared to average precipitation. All these factors combined might have contributed to the reported results in the paper. Text is added to the manuscript to clarify this. Line 268-270

Line 220: Do you consider AET as a climate variable, or a plant/crop variable (because carbon gain comes with water loss necessarily).

**RESPONSE: AET is indeed more of a crop variable as it is the direct result of plant growth which itself is driven by radiation, temperature and soil moisture. Due to the complexity of representing such relationships within a simple regression framework, we opted to leave out AET from the model fitting as discussed above.**

Fig 5: very nice figure. Interesting that there is some tail dependence for hot and dry extremes, in that in this bottom-right quadrant you see very extreme joint temp/sm anomalies compared to the others. does this raise questions of causality around the fact that such extreme low SM values can only be reached with very high Tmax? in other words, is the yield impact especially severe because of the compound impacts of temp and moisture, or simply because of extreme moisture impacts that can only happen if T is also high?

RESPONSE: Thank you. We agree with regards to the remark highlighted by the reviewer. The plot indeed hints at some tail dependence which proposes that soil moisture and temperature coupling is stronger in that bottom-right quadrant (i.e. for extreme hot-dry conditions). This likely relates to circulation and land-atmosphere feedbacks discussed previously making it particularly difficult to disentangle moisture and temperature effects during hot-dry summers. Still, we are inclined to believe that impacts are particularly severe due to the compound nature of the stress rather than it being mainly related to extreme dry conditions that only happens to occur during very high temperature periods. This answer is motivated by leaf-scale experiments showing that drought and heat inhibit plant growth via different pathways resulting in more damage to crops when these stressors occur simultaneously (Rizhsky et al., 2002, 2004; Suzuki et al., 2014).

Also could clarify in panel b that the slopes of those lines are the tmax slope + interaction slope \* 5-50-95 percentile soil moisture value. Also, given the low sample of hot and wet events, I wonder if it even makes sense to draw the blue line beyond 2sigma Tmax anomalies (there are no such events observed as you say, probably for an important climate reason).

**RESPONSE:** We added the calculation clarification to the text and adjusted the plot to limit it to physically plausible ranges. Line 347-348 in combination with Line 365.**

Line 255: there is a strong role of relatively few years in these time series, and possible some signal of climate oscillations, that may be worth at least referring to (Lesk and Anderson 2021 ERL and/or refs therein might be useful)

**RESPONSE:** We agree with regards to the strong role of relatively few years. We particularly see a high frequency of hot-dry years during the 1980s followed by a reduced frequency afterwards as reported and discussed in the referenced paper (Lesk and Anderson 2021). We added text in

**the revised manuscript to qualitatively expand on the potential role of decadal variability influencing these trends. Line 399-403.**

Line 290: pun intended?

**RESPONSE:** Not really, we used the term "lead" instead of "yield" in the revised text to avoid confusing readers. Line 460.**

Lines 300-310: It's also worth noting that Schlenker and Roberts (2009) and Schauberger et al (2017) too found that the crop damages beyond the ~30 degree threshold were mitigated when moisture was sufficient (either from irrigation or rain). So your findings are in loose agreement with those studies too, in addition to Carter et al. (2016), Siebert et al. (2017), and Troy et al. (2015). I also think it's worth acknowledging that wet conditions may simply prevent very high temperatures, thus reducing exposure rather than sensitivity to heat (see my comments on Fig. 5).

**RESPONSE:** We included these references and added the suggested nuance to the revised text. Line 370-372.**

Lines 309-311: I have a paper in review showing evidence for this globally. If it is accepted in time, it would be a good reference.

**RESPONSE: We will keep an eye on that. Thanks!**

Lines 311-313: Again, I think you're overstating the lack of attention a bit, see suggested refs above.

**RESPONSE:** All related statements have been toned down or removed from the revised manuscript.**

Nice work thanks!

**RESPONSE:** Thank you, very insightful review!**

**References:**

Ben-Ari, T., Boé, J., Ciais, P., Lecerf, R., Van Der Velde, M. and Makowski, D.: Causes and implications of the unforeseen 2016 extreme yield loss in the breadbasket of France, Nat. Commun., 9(1), doi:10.1038/s41467-018-04087-x, 2018.

Butler, E. E. and Huybers, P.: Adaptation of US maize to temperature variations, Nat. Clim. Chang., 3(1), 68–72, doi:10.1038/nclimate1585, 2013.

Carter, E. K., Melkonian, J., Riha, S. J. and Shaw, S. B.: Separating heat stress from moisture stress: Analyzing yield response to high temperature in irrigated maize, Environ. Res. Lett., 11(9), doi:10.1088/1748-9326/11/9/094012, 2016.

Carter, E. K., Melkonian, J., Steinschneider, S. and Riha, S. J.: Rainfed maize yield response to management and climate covariability at large spatial scales, Agric. For. Meteorol., 256–257(March), 242–252, doi:10.1016/j.agrformet.2018.02.029, 2018.

Cucchi, M., Weedon, G. P., Amici, A., Bellouin, N., Lange, S., Schmied, M., Hersbach, H. and Buontempo, C.: WFDE5: bias adjusted ERA5 reanalysis data for impact studies, Prep., (April), 1–32, doi:10.5194/essd-2020-28, 2020.

Gu, L., Hanson, P. J., Post, W. M. A. C. and Dale, P.: The 2007 Eastern US Spring Freeze : Increased Cold Damage in a Warming World ?, , 58(3), 2008.

James, G., Witten, D., Hastie, T. and Tibshirani, R.: An Introduction to Statistical Learning, 1st ed., Springer New York, New York, NY., 2013.

Martens, B., Miralles, D. G., Lievens, H., Van Der Schalie, R., De Jeu, R. A. M., Fernández-Prieto, D., Beck, H. E., Dorigo, W. A. and Verhoest, N. E. C.: GLEAM v3: Satellite-based land evaporation and

root-zone soil moisture, Geosci. Model Dev., 10(5), 1903–1925, doi:10.5194/gmd-10-1903-2017, 2017.

Meyer, D. W. and Badaruddin, M.: Frost tolerance of ten seedling legume species at four growth stages, Crop Sci., 41(6), 1838–1842, doi:10.2135/cropsci2001.1838, 2001.

Mourtzinis, S., Specht, J. E. and Conley, S. P.: Defining Optimal Soybean Sowing Dates across the US, Sci. Rep., 9(1), 1–7, doi:10.1038/s41598-019-38971-3, 2019.

Ortiz-Bobea, A., Wang, H., Carrillo, C. M. and Ault, T. R.: Unpacking the climatic drivers of US agricultural yields, Environ. Res. Lett., 14(6), doi:10.1088/1748-9326/ab1e75, 2019.

Portmann, F. T., Siebert, S. and Döll, P.: MIRCA2000-Global monthly irrigated and rainfed crop areas around the year 2000: A new high-resolution data set for agricultural and hydrological modeling, Global Biogeochem. Cycles, 24(1), n/a-n/a, doi:10.1029/2008gb003435, 2010.

Rizhsky, L., Liang, H. and Mittler, R.: The combined effect of drought stress and heat shock on gene expression in tobacco, Plant Physiol., 130(3), 1143–1151, doi:10.1104/pp.006858, 2002.

Rizhsky, L., Liang, H., Shuman, J., Shulaev, V., Davletova, S. and Mittler, R.: When defense pathways collide. The response of arabidopsis to a combination of drought and heat stress 1[w], Plant Physiol., 134(4), 1683–1696, doi:10.1104/pp.103.033431, 2004.

Suzuki, N., Rivero, R. M., Shulaev, V., Blumwald, E. and Mittler, R.: Abiotic and biotic stress combinations, New Phytol., 203(1), 32–43, doi:10.1111/nph.12797, 2014.

Tack, J., Barkley, A. and Hendricks, N.: Irrigation offsets wheat yield reductions from warming temperatures, Environ. Res. Lett., 12(11), doi:10.1088/1748-9326/aa8d27, 2017.

Troy, T. J., Kipgen, C. and Pal, I.: The impact of climate extremes and irrigation on US crop yields, Environ. Res. Lett., 10(5), doi:10.1088/1748-9326/10/5/054013, 2015.

Zhu, X. and Troy, T. J.: Agriculturally Relevant Climate Extremes and Their Trends in the World's Major Growing Regions, Earth's Futur., 6(4), 656–672, doi:10.1002/2017EF000687, 2018.

**Referee 2**

Thank you for the opportunity to review this paper.

The study by Hamed et al. investigates the effect of growing season hydro-climatic conditions, including hot-dry compound extremes, on US soybean yield variability. In a first step, the authors identify a set of most important climate and hydrological predictors that affect soybean yield variability across the US. In a second step, they fit statistical models to county-level yield time series to examine the strength and direction of the relationship between hydro-climatic predictors and yield outcomes. In particular, the study finds that the co-occurrence of hot and dry events leads to more negative yield outcomes than the effect of hot or dry conditions alone would predict. The authors finally investigate the effect of historical hydro-climatic trends on soy yields. The authors show that historically, there have been wetting and cooling trends across important production regions in the US. However, in the same regions, compound hot-dry extremes increased in frequency. These results highlight that the effect of compound events may be masked when looking at statistical relationships of individual variables alone, without considering interactions between hydro-climatic extremes.

The paper is clearly and well written. From my perspective the manuscript is largely suitable for publication as it stands. I only have a few suggestions for the authors to consider which will hopefully help improve this paper for publication.

**RESPONSE:** We thank the reviewer for the positive feedback on our manuscript. We are grateful for suggestions to improve our manuscript. We respond to the comments given in the text below (in bold and italics text).

**General comments:**

Overall, the statistical approach is robust, and limitations are clearly presented in the text. However, I would ask the authors to consider the following suggestions:

**1) Predictor selection**

The authors apply a strict predictor selection process, which eliminates the occurrence of highlycorrelated predictors – both at the same time as well as in subsequent months.

However, I wonder whether this approach eliminates predictors that do have an important effect on soy yields. The example presented in the text is: "we excluded soil moisture in September as August soil moisture was already selected". I understand the reasoning to avoid collinearity, but it appears a little arbitrary – likely soil moisture would be relevant in both August and September (and potentially across the whole season).

RESPONSE: We agree with this point that was also highlighted by reviewer 1. In order to avoid arbitrarily selecting from collinear predictors, we adapted the methodology to no longer intervene manually with predictor selection and only monitor multicollinearity concerns using the variance inflation factor (VIF). In the latter case, a flag is raised if the VIF exceeds a value of 3 for any variable used to fit the final model at county scale (Carter et al., 2016; James et al., 2013).

Would it be more suitable to consider three aggregations for each variable (monthly, seasonal and the whole growing season) and select only one temporal aggregation per predictor in the final model? In this configuration, a predictor of "growing season soil moisture" could have been selected by the algorithm, if it was found to have the highest correlation with yields. This would lead to more interpretable results in the context of understanding climate influences on soy yields.

**RESPONSE:** We understand this concern and therefore ran a test with the suggested modification.**

As an initial disclaimer, the general methodology has been adapted to run the selection process and model fitting at county level as this was a particular concern for reviewer 1. This implies that different counties can now have a different set of predictors. In addition, the methodology was adapted to exclude the manual selection step as stated in the response above. To limit the number of potential predictors to select from, we reduced the initial set of considered variables to only include Minimum Temperature, Maximum Temperature, Root Zone Soil Moisture and Excessive precipitation. These predictors are supported by main findings in prior literature that highlights the damaging effects of chilling conditions, high temperature, water stress and excessive rainfall on crops grown in the US (Carter et al., 2018; Gu et al., 2008; Li et al., 2019; Mourtzinis et al., 2015, 2019; Ortiz-Bobea et al., 2019; Zipper et al., 2016). We refer the reviewer to the response to RC1 for more details on the change in the methodology. An adapted overview of the overall modelling workflow is presented in Fig. 2.

Back to the initial point, the reviewer suggested here a different selection approach where one temporal aggregation per predictor is selected in the final model. The main premise was to allow for growing season length predictors to be selected by the model if these were found to be most suitable to explain local soy yield variability. Consequently, for this exercise, we modified our selection approach to consider growing season predictors and to select one best temporal aggregation per predictor rather than two best moisture and temperature related variables for each distinct period of the growing season (i.e. early-, mid- and late). Results showed that the full growing season temporal aggregation was only picked up on very few occasions and only minimum temperature in northern states showed a clear signal for growing season length predictors (Figure R1). We believe that this can be explained by the changing sensitivity of soy crop yields to climatic variables across the season. For instance, warm temperature generally

increases soy yields in early and late season but is associated to important reductions during the mid-season. Furthermore, short-term damaging conditions coinciding with particularly vulnerable stages of the crop growth cycle can result in important yield losses (Ben-Ari et al., 2018; Carter et al., 2018; Tack et al., 2017; Troy et al., 2015). Full season averages can mask out such details. The masking out effect can explain why full season predictors were seldomly picked-up throughout the adapted selection approach. Figure R3 displays general model performance. The model that contains season average predictors performs qualitatively very similar to our initial setup. To keep the focus within this manuscript on the importance of timing with regards to soybean yield climate sensitivities, we prefer to keep this aspect of the method similar to what we initially proposed in the preprint and avoid the inclusion of seasonal averages. We added a sentence in the revised manuscript saying that we've tested the inclusion of seasonal averages and these were not found to be critical for our setup. Line 125-127

Figure R1. Region- and season- specific selected temperature and moisture related predictors when selecting for one best temporal aggregation per predictor.

---

## Referee Report (RR1)

A big thanks to the authors for the thorough revisions. I think the paper is basically ready to be accepted, but a few things may need to be clarified.

1) The new analysis on sub-seasonal correlations among tmax, sm, and aet are very interesting and add a lot to the paper, so thanks to the authors for doing this. I think the method could be clarified: I'm not totally clear what 'monthly correlations' means. I'm pretty sure it is interannual correlation between pairs of variables for a given month of the year, repeated over the various months. If so, this is a reasonably fair way to quantify these couplings, but the method should be clarified, and the rationale expanded a bit (and/or simply cite Seneviratne et al. 2010). Also, does this calculation incorporate data from all the counties into the correlation?

2) I don't quite follow the claim based on Fig 4 on line 330 that soybean in the south is particularly sensitive to combined high summer temperature and low fall moisture. While the spatial pattern you describe is generally right, many counties are different between those two maps. Also, the combination of positive fall soil moisture coefficients and negative summer temperature coefficients doesn't necessarily mean 'compound impacts' – these could be just independent sensitivities. Further, the signal could be driven by different years in the regression model – i.e. dry falls some years driving the coefficient negative, hot summers in other driving that coefficient negative. I think it would be stronger to refer to Fig. A3, as significant interactions terms are more direct support for this 'compounding' effect, and a better link into Fig. 5 which really shows this compounding in detail (based on interaction terms).

3) Fig. 4 caption is a bit confusing. The interaction coefficients are shown in Fig. A3, right, but not here? So perhaps drop the last sentence. And could explain more which among the variables in Fig. 3a are being shown in – e.g. for temperature-related, this is *either* Tmax or Tmin, depending on which one dominates in a given county, right? And could refer to Figs A1-2 to help reader find these.

A couple other interesting things you could include:

One point that to me supports and ties in the new coupling analysis is that, in the updated results, there is a sharp boundary between positive and negative yield impact of mid-season temperature variables, and this boundary (somewhere in the middle of Iowa) seems curiously close to the transition from energy- to moisture-limited summer soil moisture regime in North America (see Seneviratne et al. 2010). You could mention that as it supports part of your interpretation of Figure 7. In my own work I also find a global pattern consistent with this (stronger T-ET coupling worsens yield sensitivity to temperature for soybean) in the paper I mentioned during my first review (which is out now by the way, Lesk et al. 2021)

The August Tmax – Sept. Tmin interaction is also very interesting and indeed may be enhanced sensitivity to cold from heat acclimation in the early season. You could expand on what physiologically might cause this (not so obvious, at least to me).

References:

Seneviratne, S. I., Corti, T., Davin, E. L., Hirschi, M., Jaeger, E. B., Lehner, I., ... & Teuling, A. J. (2010). Investigating soil moisture–climate interactions in a changing climate: A review. *Earth-Science Reviews*, *99*(3-4), 125-161.

Lesk, C., Coffel, E., Winter, J., Ray, D., Zscheischler, J., Seneviratne, S. I., & Horton, R. (2021). Stronger temperature–moisture couplings exacerbate the impact of climate warming on global crop yields. *Nature Food*, *2*(9), 683-691.

---

## Author Response (AR2)

*Dear Editor,*

*Thank you for the opportunity to potentially publish our manuscript within this special issue on "Understanding compound weather and climate events and related impacts". We would like to thank again the reviewers for their helpful and constructive comments. We were pleased to read the reviewers support publication provided that some minor comments are dealt with. In what follows, we expand on the remaining points left to be clarified. Referenced Line numbers refer to the updated track changed version of the manuscript. Given the small amount of reviewer comments for this iteration, we also added the changes made in the text to this document in addition to providing a marked-up manuscript version showing the changes.*

**Point-by-point response to the reviewers:**
**Referee 1: Corey Lesk**

A big thanks to the authors for the thorough revisions. I think the paper is basically ready to be accepted, but a few things may need to be clarified.

*Thank you for the detailed revisions which helped considerably improve the quality of our manuscript. We are glad the paper is perceived almost ready to be accepted. In what follows, we expand on the remaining points left to be clarified.*

1) The new analysis on sub-seasonal correlations among tmax, sm, and aet are very interesting and add a lot to the paper, so thanks to the authors for doing this. I think the method could be clarified: I'm not totally clear what 'monthly correlations' means. I'm pretty sure it is interannual correlation between pairs of variables for a given month of the year, repeated over the various months. If so, this is a reasonably fair way to quantify these couplings, but the method should be clarified, and the rationale expanded a bit (and/or simply cite Seneviratne et al. 2010). Also, does this calculation incorporate data from all the counties into the correlation?

*Thank you. What is meant is indeed interannual correlation between pairs of variables for a given month of the year, repeated over the various months of the year. We adjusted the text in the method section accordingly (see Line 175-179), in the result section (Line 342-343), the caption of Figure 7, the discussion section (Line 446-449) and the caption for Figure A6. This calculation incorporates data from all counties into the correlation by first spatially averaging single variables over the entire rainfed harvested area (See Figure 1) and then quantifying the couplings. It can be argued that by including all counties, we even underestimated the effect at specific locations. For instance, if we would have focused only on counties that highlight hot-dry sensitivities in the regression model, we would have obtained even higher values. Figure A6 depicts how interannual correlation between AET and Tmax vary at the grid-cell level. Nevertheless, what is currently presented in this manuscript can be considered a conservative estimate of these couplings that can be further explored in future research.*

*Manuscript changes:*

*"Coevolution of considered variables was estimated by calculating the interannual correlation between pairs of variables for a given month of the year, repeated over the various calendar months (Seneviratne et al., 2010). This calculation incorporates data from all counties into the correlation by first spatially averaging single variables over the entire rainfed harvested area (Fig. 1) and then quantifying the couplings. Moreover, we calculated correlation at the grid-cell level between actual evapotranspiration and maximum temperature to show how these couplings can differ at the local scale."*

*"Interannual correlation between root-zone soil-moisture (SMroot), maximum temperature (Tmax) and actual evapotranspiration (AET) pairs for a given month of the year, repeated over the various calendar months are used to estimate the coupling strength during hot-dry summer years and normal summer years."*

*Figure 7. Interannual correlation between various pairs of temperature and moisture variables for a given month of the year, repeated over the various calendar months conditioned on hot-dry events. Dots indicate statistical significance at the 95% confidence. Shaded regions represent important differences in the couplings that can play a critical role in the development of hot-dry events.*

*"We illustrated this mechanism by analyzing the evolution of land-atmosphere coupling within the growing season, captured by interannual correlations between root-zone soil-moisture (SMroot), maximum temperature (Tmax) and actual evapotranspiration (AET) pairs for a given month of the year, repeated over the various calendar months. "*

*"Figure A 6. Interannual correlation between actual evapotranspiration and maximum temperature for a given month of the year conditioned on hot-dry events and repeated for the period going from June to September. Dots indicate statistical significance at the 95% confidence. "*

2) I don't quite follow the claim based on Fig 4 on line 330 that soybean in the south is particularly sensitive to combined high summer temperature and low fall moisture. While the spatial pattern you describe is generally right, many counties are different between those two maps. Also, the combination of positive fall soil moisture coefficients and negative summer temperature coefficients doesn't necessarily mean 'compound impacts' – these could be just independent sensitivities. Further, the signal could be driven by different years in the regression model – i.e. dry falls some years driving the coefficient negative, hot summers in other driving that coefficient negative. I think it would be stronger to refer to Fig. A3, as significant interactions terms are more direct support for this 'compounding' effect, and a better link into Fig. 5 which really shows this compounding in detail (based on interaction terms).

*We agree with the reviewer and have therefore adjusted the reference on line 260 (330 in the previously submitted manuscript including track changes) from Fig 4 to Fig. A3.*

3) Fig. 4 caption is a bit confusing. The interaction coefficients are shown in Fig. A3, right, but not here? So perhaps drop the last sentence. And could explain more which among the variables in Fig. 3a are being shown in – e.g. for temperature-related, this is either Tmax or Tmin, depending on which one dominates in a given county, right? And could refer to Figs A1-2 to help reader find these.

*We agree with the reviewer and have therefore adjusted the caption for Fig.4 (Line 231)*

*Manuscript changes:*

*"Figure 4. Region- and season-specific estimated sensitivity coefficients for soybean yield and selected predictors. Stippling indicates statistical significance from a t-test at 95% confidence level. Values of coefficients are interpreted as the change in soybean yield standard deviation from a one-standard deviation change in the considered independent variable. Temperature-related variables can refer to either Tmin or Tmax depending on the selected variable in a given county. Similarly, moisture-related variable can refer to either SMroot or Num_pr20. Finally, for each seasonal bracket (i.e. Early, Mid or Late), the selected time resolution for each variable can be either a seasonal aggregate or the value for a specific month within that bracket. We refer the reader to the appendix (Fig. A1, A2) for a more detailed account of selected variables per county."*

A couple other interesting things you could include: One point that to me supports and ties in the new coupling analysis is that, in the updated results, there is a sharp boundary between positive and negative yield impact of mid-season temperature variables, and this boundary (somewhere in the middle of Iowa) seems curiously close to the transition from energy- to moisture-limited summer soil moisture regime in North America (see Seneviratne et al. 2010). You could mention that as it supports part of your interpretation of Figure 7. In my own work I also find a global pattern consistent with this (stronger T-ET coupling worsens yield sensitivity to temperature for soybean) in the paper I mentioned during my first review (which is out now by the way, Lesk et al. 2021)

***Thank you for highlighting this. We've added the following text (Line 426-430) to the discussion to enhance result interpretation:***

***Manuscript changes:***

***[…] "These regions are characterized by a predominantly energy-limited summer regime where the role of soil moisture in related land-atmosphere feedbacks is limited (Seneviratne et al., 2010). These northern states also showed reduced sensitivity to high temperatures over summer (Fig. 4d) in line with Lesk et al. (2021b) who highlighted reduced soybean yield sensitivity to temperature in energy-limited regimes at the global scale."***

The August Tmax – Sept. Tmin interaction is also very interesting and indeed may be enhanced sensitivity to cold from heat acclimation in the early season. You could expand on what physiologically might cause this (not so obvious, at least to me).

***We agree that this result is interesting although we limited the discussion of this topic as the paper focuses on hot-dry conditions. Still, we adjusted the following text (Line 259-260) to clarify the physiological process that is linked to crop acclimation.***

***Manuscript changes:***

***"This might reflect increased impacts whenever anomalously hot conditions in peak summer are followed by anomalously cold conditions in September-October. Optimal temperature for crop photosynthesis fluctuates due to the capacity of the crop to seasonally adjust its physiological response to temperature (Kumarathunge et al., 2019). It follows that consistent high temperature within the growing season can make the crops more productive at higher temperatures. The abrupt change in temperature conditions from hot to cold further stresses crops and reduces the potential positive effects of crop temperature acclimation (Butler and Huybers, 2013; Carter et al., 2016)."***

References: Seneviratne, S. I., Corti, T., Davin, E. L., Hirschi, M., Jaeger, E. B., Lehner, I., ... & Teuling, A. J. (2010). Investigating soil moisture–climate interactions in a changing climate: A review. Earth-Science Reviews, 99(3-4), 125-161. Lesk, C., Coffel, E., Winter, J., Ray, D., Zscheischler, J., Seneviratne, S. I., & Horton, R. (2021). Stronger temperature–moisture couplings exacerbate the impact of climate warming on global crop yields. Nature Food, 2(9), 683-691.

Butler, E. E. and Huybers, P.: Adaptation of US maize to temperature variations, Nat. Clim. Chang., 3(1), 68–72, doi:10.1038/nclimate1585, 2013.

Carter, E. K., Melkonian, J., Riha, S. J. and Shaw, S. B.: Separating heat stress from moisture stress: Analyzing yield response to high temperature in irrigated maize, Environ. Res. Lett., 11(9), doi:10.1088/1748-9326/11/9/094012, 2016.

Kumarathunge, D. P., Medlyn, B. E., Drake, J. E., Tjoelker, M. G., Aspinwall, M. J., Battaglia, M., Cano, F. J., Carter, K. R., Cavaleri, M. A., Cernusak, L. A., Chambers, J. Q., Crous, K. Y., De Kauwe, M. G., Dillaway, D. N., Dreyer, E., Ellsworth, D. S., Ghannoum, O., Han, Q., Hikosaka, K., Jensen, A. M., Kelly, J. W. G., Kruger, E. L., Mercado, L. M., Onoda, Y., Reich, P. B., Rogers, A., Slot, M., Smith, N. G., Tarvainen, L., Tissue, D. T., Togashi, H. F., Tribuzy, E. S., Uddling, J., Vårhammar, A., Wallin, G., Warren, J. M. and Way, D. A.: Acclimation and adaptation components of the temperature dependence of plant photosynthesis at the global scale, New Phytol., 222(2), 768–784, doi:10.1111/nph.15668, 2019.

---

## Author Response (AR3)

Dear Editor,

Thank you for your comment. We are grateful the manuscript can now be accepted subject to technical corrections. We've adjusted the mentioned reference and double checked all other references in the manuscript. We've enjoyed this submission and review process and thank all parties involved.

Best regards,
Raed Hamed, on behalf of all authors.